# Object-Centric Concept-Bottlenecks

David Steinmann[1,2]    Wolfgang Stammer[1,2]    Antonia Wüst[1]
Kristian Kersting[1,2,3,4]

[1]Computer Science Department, TU Darmstadt; [2]Hessian Center for AI (hessian.AI);
[3]German Research Center for AI (DFKI); [4]Centre for Cognitive Science, TU Darmstadt

## Abstract

Developing high-performing, yet interpretable models remains a critical challenge in modern AI. Concept-based models (CBMs) attempt to address this by extracting human-understandable concepts from a global encoding (*e.g.*, image encoding) and then applying a linear classifier on the resulting concept activations, enabling transparent decision-making. However, their reliance on holistic image encodings limits their expressiveness in object-centric real-world settings and thus hinders their ability to solve complex vision tasks beyond single-label classification. To tackle these challenges, we introduce Object-Centric Concept Bottlenecks (OCB), a framework that combines the strengths of CBMs and pre-trained object-centric foundation models, boosting performance and interpretability. We evaluate OCB on complex image datasets and conduct a comprehensive ablation study to analyze key components of the framework, such as strategies for aggregating object-concept encodings. The results show that OCB outperforms traditional CBMs and allows one to make interpretable decisions for complex visual tasks.

## 1 Introduction

In recent years, the field of interpretable machine learning has made significant progress, particularly through the development of interpretable-by-design models (Chen et al., 2019; Koh et al., 2020; Rudin et al., 2021; Chattopadhyay et al., 2023b,a). These models are designed to provide explanations that faithfully reflect the model's internal reasoning, thereby improving user understanding and control. One of the most prominent lines of research in this area is concept bottleneck models (CBMs) (Koh et al., 2020; Stammer et al., 2021), which aim to ground a model's internal representations in semantically meaningful high-level concepts. The core idea is to decompose complex inputs, such as images, into an interpretable concept encoding and to make predictions based on this concept-level abstraction. To improve the practical applicability of this approach, many recent CBM methods have leveraged the power of pretrained models, reducing the amount of required annotations and prior knowledge (Oikarinen et al., 2023; Yang et al., 2023; Bhalla et al., 2024; Yamaguchi et al., 2025).

While such recent advances have enhanced CBMs, their deployment has thus far been largely confined to simpler tasks, such as single-label image classification. One reason for this limitation is that CBMs typically extract concepts from an entire image, leading to a single, holistic encoding. However, addressing more complex visual reasoning tasks (*e.g.*, as illustrated in Fig. 1) requires a shift towards object-based processing. Objects provide a natural and intuitive abstraction for visual reasoning, as decomposing complex scenes into discrete, meaningful components simplifies the analysis of structure and relations. Importantly, object-centric representations also enhance transparency by supporting more detailed, human-aligned explanations.

Despite this, object-centricity has largely been overlooked in recent CBM research, particularly in natural image domains. While earlier concept-based models working with synthetic data have successfully integrated object-centric representations (Stammer et al., 2021, 2024b; Delfosse et al., 2024), most SOTA CBM approaches for natural images continue to rely on image-level encodings.

39th Conference on Neural Information Processing Systems (NeurIPS 2025).

In this work, we aim to bridge this gap by integrating the advantages of object-centric modeling into CBMs, with a focus on models that can handle complex, natural images while maintaining interpretability through structured, object-level representations.

To this end, we introduce Object-Centric Concept Bottlenecks (OCB). In this framework, an object proposal module first identifies relevant objects in an image, which are then represented by high-level concepts from a concept discovery module. For the final task prediction, concept activations from both the original image and individual objects are *aggregated* and passed through a linear prediction layer. Following recent trends in interpretable-by-design research, OCB utilizes pretrained models for both object detection and concept discovery, thereby reducing the amount of required human supervision. As the addition of object-level representations expands the concept space, effectively aggregating these becomes a new challenge. Thus, our evaluations include a detailed analysis of the critical components of OCB. We conduct experiments across multiple datasets, including our newly introduced COCOLogic benchmark, based on MSCOCO (Lin et al., 2014), which requires models to classify images according to concepts defined by logical combinations of objects. Importantly, we also evaluate CBMs via OCB for the first time on multi-label image classification benchmarks.

**Image-level Concept Reasoning**

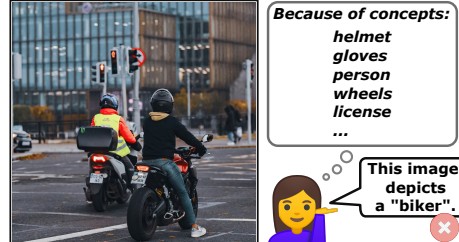

**Object-level Concept Reasoning**

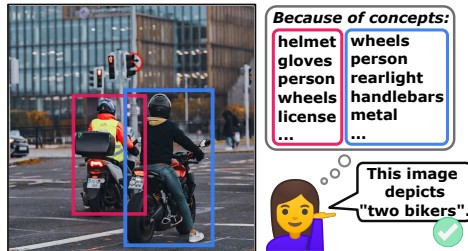

Figure 1: Reasoning and explaining on the object-level requires object representations.

Overall, our work shows that Object-Centric Concept Bottlenecks represents an important step towards more competitive, yet interpretable AI models, with a particular focus on object-centricity and its application to CBMs for natural images. Our contributions can be summarised as:

(i) Introducing object-level representations into the framework of CBMs.

(ii) Extending concept-based models for the first time to more complex classification settings, including multi-label classification.

(iii) Providing a detailed performance analysis of OCB's components, *e.g.*, aggregation strategies.

(iv) Introducing COCOLogic, a new benchmark dataset for complex object-based classifications of real-world images.

The remainder of the paper is structured as follows. We begin with a review of related work, highlighting recent developments in the field. We then introduce our proposed framework, Object-Centric Concept Bottlenecks, and present its formal description. This is followed by a comprehensive experimental evaluation, including several ablation studies that examine key components of the framework. Finally, we discuss our findings and conclude the paper.[1]

## 2 Related Work

**Concept-Based Models.** The introduction of Concept Bottleneck Models (CBMs) (Koh et al., 2020) marked an important moment in the growing interest in concept-based models (*cf.* (Yeh et al., 2021; Fel et al., 2023) for overviews on concept-based explainability). Their appeal lies in the promise of interpretable predictions and a structured interface for human interaction. While the original CBM framework relied on fully supervised concept annotations, subsequent research has relaxed this requirement by introducing unsupervised concepts (Sawada and Nakamura, 2022), leveraging pretrained vision-language models like CLIP for concept extraction (Bhalla et al., 2024; Yang et al., 2023; Oikarinen et al., 2023; Panousis et al., 2024; Yamaguchi et al., 2025), or employing fully unsupervised concept discovery methods (Stammer et al., 2024b; Schrodi et al., 2024; Schut et al., 2025). Efforts have also focused on improving the bottleneck interface by mitigating concept

---

[1]Code and data available at: https://github.com/DavSte13/Object-Centric-Concept-Bottlenecks/

leakage (Havasi et al., 2022) and enabling dynamic expansion of the concept space (Shang et al., 2024). Despite this progress, prior work has not explored the potential of explicit object-centric representations. Although some studies have incorporated such representations into bottleneck models (Yi et al., 2018; Stammer et al., 2021; Wüst et al., 2024; Delfosse et al., 2024), these approaches have thus far been limited to synthetic data. In contrast, our work introduces object-centric bottlenecks for real-world image data. While Prasse et al. (2025) utilizes foundation models to extract an object-based concept bank from images, they compute a single concept activation vector per image, in contrast to OCB, which explicitly combines image and object-level concept activations.

**Object-Centric Representations.** Object-centric representations (decomposing scenes into discrete, object-based components) have emerged as a powerful inductive bias, facilitating compositional reasoning, transferability, and sample-efficient learning across domains such as robotics (Shi et al., 2024), video understanding (Tang et al., 2025), and vision-language tasks (Assouel et al., 2025; Didolkar et al., 2025; Karimi-Mamaghan et al., 2025). Early slot-based approaches demonstrated the promise of object-centric learning in synthetic settings (Eslami et al., 2016; Burgess et al., 2019; Lin et al., 2020; Greff et al., 2019; Locatello et al., 2020), while region-based models, such as the R-CNN family (Girshick, 2015; He et al., 2017), established object-level reasoning through supervised detection and segmentation. More recent developments, including GENESIS-V2 (Engelcke et al., 2021) and DINOSAUR (Seitzer et al., 2023), have extended slot-based decomposition to natural images, making object-centricity increasingly practical for real-world data. Object-centric learning also remains of interest in the context of foundation models, *e.g.*, regarding compositional generalization. For example, object-centric models outperform foundation model baselines in compositional visual question answering tasks Kapl et al. (2025); Carion et al. (2020). Furthermore, recent large-scale models such as SAM (Kirillov et al., 2023) and DeiSAM (Shindo et al., 2024) adopt object-level interfaces, underlining the growing role of structured, object-aware representations in scalable AI systems. Following this trend, OCB integrates object-centric representations into interpretable-by-design models, aiming to improve both predictive performance and model transparency.

**Utilizing Pre-trained Models for Interpretability.** A growing body of work investigates how large pretrained models, such as vision-language models (VLMs) and large language models (LLMs), can be leveraged to enhance interpretability, particularly through concept bottlenecks. Many recent approaches incorporate features from pretrained models into inherently interpretable architectures to avoid the need for manual concept supervision. For instance, Yang et al. (2023) and Rao et al. (2024) extract concepts from language models or detection backbones, while Ismail et al. (2024) introduce CB-pLMs for controllable protein design. In natural language processing, Tan et al. (2024) retrofit LLMs with lightweight bottlenecks, and Sun et al. (2024) scale this to classification and generation with competitive performance and built-in interpretability. Further, Chen et al. (2025) improve self-explanations by aligning internal LLM representations with explicit concept subspaces. In line with these works, OCB utilizes pretrained models to build object-centric concept representations.

## 3 Object-Centric Concept Bottlenecks (OCB)

In this section, we introduce our novel framework Object-Centric Concept Bottlenecks (OCB). It consists of three different components: (I) an object proposal module that detects and crops relevant objects from an image, (II) a concept discovery module that transforms images into a set of human-understandable concepts and (III) a predictor module that solves the task based on theses concepts in an interpretable fashion. With that, OCB provides a competitive and interpretable architecture to handle complex visual tasks in an object-centric way. Before going into the details of the individual components, we first introduce the necessary background notation.

**Background.** To build a general concept-based model, let us assume we have access to some image data $\mathcal{X} \in \mathbb{R}^{N \times D}$, which consists of $N$ images of dimension $D$. Additionally, we have some labels $\mathcal{Y} \in \mathbb{R}^{N \times M}$, for example, class or category labels for multiclass or multilabel classification. We want to develop a concept-based model $f$ that predicts the labels given the input $f : \mathcal{X} \to \mathcal{Y}$, with $f(x) = \hat{y} \in \mathbb{R}^M$ for a given input $x$. For an interpretable prediction, CBMs commonly split $f$ into two stages: predicting human-understandable concepts from the input and then performing the task based on these concepts. Thus, a discovery module $h : \mathcal{X} \to \mathcal{C}$ predicts the presence of high-level concepts $\mathcal{C} \in \mathbb{R}^{N \times C}$ from the input with a concept space of dimension $C$. Then, an interpretable classifier (*e.g.*, a linear model) $g : \mathcal{C} \to \mathcal{Y}$ predicts the task labels based on the concept activations from $h$. Together, $g$ and $h$ represent the concept-based model $f(x) = g(h(x))$.

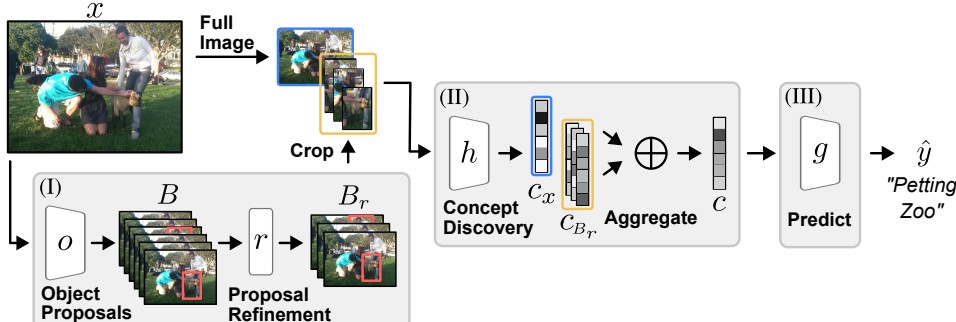

Figure 2: **Object-Centric Concept Bottlenecks** combine object-centric representations with concept-based modeling in a three-stage pipeline: (I) An object proposal module identifies and refines object candidates within an image. (II) A concept discovery module encodes the entire image and its object crops into human-understandable concept activations. (III) These activations are aggregated and passed to a simple, interpretable predictor to generate the final output. This architecture enables interpretable, object-aware reasoning for complex visual tasks.

**From Images to Objects.** To incorporate object representations into concept-based models, OCB first identifies relevant objects within an image (see (I) in Fig. 2). This is achieved using an object proposal model $o : \mathcal{X} \rightarrow (\mathcal{B}, \mathcal{S})$, which maps an input image $x$ to a set of bounding boxes $B$ and associated certainty scores $S$, indicating the confidence that each box contains a meaningful object. Since $o$ may generate numerous proposals of varying quality, OCB filters these object proposals to remove noisy object proposals. Bounding boxes that are too small (low resolution and less likely to carry meaningful concepts) or too large (too similar to the full image) are removed based on configurable size thresholds $t_{min}$ and $t_{max}$. Additionally, boxes with certainty scores below the threshold $t_{cer}$ are discarded. As many object-proposal models have a tendency to generate multiple bounding boxes for the same object, all object proposals with a high Intersection over Union (IoU) $> t_{\text{IoU}}$ are filtered out, inspired by non-maximum suppression (Neubeck and Van Gool, 2006). For this, OCB first orders the object-proposals according to their certainty scores and removes all bounding boxes that have an IoU higher than $t_{\text{IoU}}$ with any remaining bounding boxes of a higher certainty (presented here as $\text{IoU}(b, B) > t_{\text{IoU}}$). Lastly, we restrict the number of object proposals to a maximum of $k$ bounding boxes, which can, for example, be set based on the complexity of the task and the images (*cf.* full filtering pseudocode in Alg. 1 in the appendix). Based on this filtering process, we obtain a refined set of object proposals:

$$B_r = \{b | (b, s) \in o(x) \wedge t_{min} \leq \text{size}(b) \leq t_{max} \wedge s \geq t_{cer} \wedge \text{IoU}(b, B) \leq t_{\text{IoU}}\}, |B_r| \leq k \quad (1)$$

where $\text{size}(b)$ represents the total size (in pixels) of the bounding box $b$. By applying these steps to refine the set of object proposals, OCB avoids adding too many or noisy object proposals, which, in turn, would lead to noisy and uninformative concepts in the following steps.

**From Objects to Concepts.** In the next step, OCB generates the concept activations $c$ from the input image and the refined object proposals $B_r$. Given an image $i \in \mathbb{R}^D$, the concept discovery module $h(i) = c_i \in \mathbb{R}^C$ maps that image into a vector of $C$ concept activations. To retain both global and object-centric information, we first generate concepts based on the whole image $x$, obtaining $c_x = h(x)$. Then, for each object proposal $b_i \in B_r$, we crop and resize the corresponding region $x_{b_i} = \text{resize}(\text{crop}(x, b_i)) \in \mathbb{R}^D$ and compute its concept activations via $c_{b_i} = h(x_{b_i})$ (see (II) in Fig. 2). Since the number of object proposals can vary across images, we pad with zeros as concept activations to ensure a fixed number $k$ of object-centric concept representations per image:

$$\forall i \leq k : c_{b_i} \begin{cases} h(x_{b_i}), & \text{if } |B_r| \geq i \\ \{0\}^C, & \text{otherwise} \end{cases} \quad (2)$$

Together, our enriched concept activations for an input image consist of the activations for the whole image and all object proposals: $c = \{c_x\} \cup \{c_B\} = \{c_x, c_0, \cdots, c_k\} \in \mathbb{R}^{(k+1)C}$. By default, the concept space for image-level concepts and object-level concepts is the same. If this is the case, the underlying assumption of OCB is that these concept spaces are sparsely activated, so that not every concept is active (to some extent) for every image and object.

**Aggregating Object Concept Encodings.** To enable prediction based on the discovered concepts, OCB aggregates concept activations from the full image and object proposals. A simple approach is

concatenation, which preserves per-object information but scales linearly with the number of objects $k$, limiting scalability. Next to (i) `concatenating` the encodings, we further suggest the use of the aggregation forms (ii) `sum`, (iii) `max`, (iv) `count`, and (v) `sum + count`. In the following, we use subscripts to indicate the object-proposal (or main image) from which a concept activation is obtained and a superscript $l$ to denote the index of the concept in the activation space.

The `max` aggregation takes the element-wise maximum over the concept-activations: $\mathtt{max}(c)^l = \arg\max_{j \in (x,0,\cdots,k)} c_j^l$. This maintains the same value range for all activations but loses information about activation strength or number. The `sum` aggregation maintains the information about the individual activation strengths: $\mathtt{sum}(c)^l = \sum_{j \in (x,0,\cdots,k)} c_j^l$. However, this aggregation still does not have explicit information about the number of times a concept has been activated, and thus about counts of objects in the image. The `count` aggregation is explicitly suited for situations where numbers of objects are important: $\mathtt{count}(c)^l = \sum_{j \in (x,0,\cdots,k)} \mathbf{I}(c_j^l)$, where $\mathbf{I}(a)$ equals 1 if $a \neq 0$ and 0 otherwise. Compared to `concat`, all these aggregations have the advantage that the output space remains $\mathbb{R}^C$, independent of the number of objects that concept activations are computed for. However, as all of these aggregations also lose some kind of information, the final aggregation method we introduce is `sum + count`. This aggregation combines `sum` and `count`: $\mathtt{sum + count}(c)^l = (\mathtt{sum}(c)^l, \mathtt{count}(c)^l)$, with an output space of $\mathbb{R}^{2C}$. This aggregation retains information about the concept activation strength as well as explicitly keeping track of concept activation counts, while still having an output space that is constant in size relative to the number of objects. We provide evaluations on the different forms of aggregations in Sec. 5.

**From Object Concepts to Decisions.** After the concept activations from the original image and the object proposals have been combined with one of the aggregations above, the last step is to compute a final prediction based on $c$ ((III) in Fig. 2). This is done with a predictor $g(\cdot)$ where we follow the typical setup of concept-bottleneck models and use a linear layer as a predictor due to its inherent interpretability. The input space of $g$ is either $\mathbb{R}^C$ (for `max`, `sum` and `count`), $\mathbb{R}^{2C}$ (for `sum + count`), or $\mathbb{R}^{(k+1)C}$ (for `concat`), depending on the aggregation method. Overall, this leads to the joint processing of OCB as shown in Fig. 2: Finding and refining object proposals (I), computing and aggregating concept activations (II) and finally predicting the task output (III).

**Utilizing Pretrained Models.** OCB relies on two core components: the object proposal model $o$ and the concept discovery module $h$. While both can be trained from scratch, doing so typically requires extensive annotations. Instead, OCB leverages pre-trained models for both, eliminating the need for costly supervision. In this setup, only the predictor network $g$ is trained, while $o$ and $h$ remain fixed.

## 4  The COCOLogic Dataset

Previous CBM research has primarily focused on simple single-label classification tasks, often using datasets where success depends largely on detecting the presence of a specific object type. In contrast, our work extends the evaluation of CBMs to both more complex multi-label settings and, for the first time, to a novel single-label classification task that demands richer reasoning capabilities on natural images. For this, we introduce COCOLogic, a new benchmark based on the MSCOCO dataset (Lin et al., 2014), that is designed to evaluate a model's ability to perform structured visual reasoning in a single-label classification context.

COCOLogic is intended as a proxy task for structured reasoning in vision, capturing aspects such as multi-object composition, negation, and hierarchical categories while still being grounded in real-world data. Concretely, COCOLogic requires models to classify images based on high-level semantic concepts defined by logical combinations of objects. These concepts incorporate disjunctions, conjunctions, negations, and counting constraints over detected object categories, far beyond simple object presence. Each image is assigned one

Pair of Pets

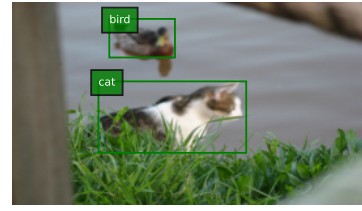

Animal Meets Traffic

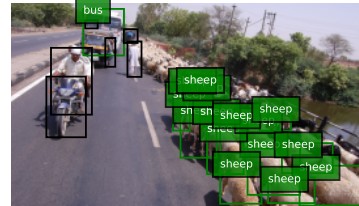

Figure 3: Two examples from the COCOLogic dataset with relevant objects for class decision.

of ten mutually exclusive labels, determined by a logical rule applied to COCO object annotations. Two examples are shown in Fig. 3 (*cf.* Sec. D for more details).

The dataset construction process ensures exclusive labeling, such that each image satisfies exactly one class definition. This setting allows for a controlled comparison of model expressiveness on complex visual reasoning tasks, while still formulated in the context of single-label classification. COCOLogic thus serves as a bridge between logical reasoning and visual understanding within a simple training setup: it remains grounded in realistic visual scenes while being inherently challenging from a logical reasoning perspective.

## 5  Experimental Evaluations

In our evaluations, we investigate the potential of object representations for performant, interpretable decision making via our novel OCB framework. We investigate the overall predictive performance, analyse the framework's individual components, and conduct a qualitative inspection of OCB's object-centric concept space. The evaluations are guided by the following research questions:

- **(RQ1)** Do object-centric representations allow for more competitive CBMs?
- **(RQ2)** How impactful is the choice of object extractor?
- **(RQ3)** Which aggregation strategy best supports task performance?
- **(RQ4)** How much does the number of objects affect OCB's performance?

**Datasets.** We evaluate OCB on both single- and multi-label image classification tasks. For multi-label settings, we use PASCAL-VOC (Everingham et al., 2010) and MSCOCO (Lin et al., 2014), distinguishing between low-level categories (COCO(l)) and high-level super-categories (COCO(h)). For single-label classification, we use SUN397 (Xiao et al., 2010, 2016) and our novel COCOLogic dataset (see Sec. 4).

**Metrics.** When comparing downstream performance of the investigated models on the single-label datasets, we follow recent work and report test accuracy (balanced accuracy for COCOLogic due to its strong class imbalance). For the multi-label datasets, we report mean Average Precision (mAP). In all experiments, we report average and standard deviation over 5 seeds.

**Models.** For our evaluations, we do not rely on any concept or object-level supervision but leverage the potential of pretrained models for instantiating the different components of OCB. For the object-proposal model $o$ we explore the use of the "narrow-purpose" pretrained model MaskRCNN (He et al., 2017) (denoted as RCNN), which has been trained to detect objects of the COCO (Lin et al., 2014) dataset. As a second, "general-purpose" proposal model, we revert to SAM (Kirillov et al., 2023), which has been pretained on SA-1B and, in principle, allows to segment arbitrary images. To instantiate the concept discovery module $h$, it is in principle possible to use any of the recent pretrained CBM approaches. We revert to the practical and quite general SpLiCE (Bhalla et al., 2024) approach, in particular due to its task-independent concept vocabulary. Moreover, SpLiCE can be seen as a generalized framework that subsumes a broad family of recent concept bottleneck models that leverage pretrained vision-language representations such as CLIP (Yang et al., 2023; Oikarinen et al., 2023; Panousis et al., 2024; Yamaguchi et al., 2025). This generality allows SpLiCE to serve as a representative instantiation for studying concept-based models in a unified framework. Additionally, SpLiCE generates sparse concept activations, which aid understandability and thus downstream interpretability.

We first evaluate a non-interpretable model that maps CLIP image embeddings to class predictions via a two-layer MLP, serving as an upper bound on performance using CLIP features alone. For interpretable baselines, we use a SpLiCE-based CBM, where a linear classifier predicts multiclass or multilabel outputs from whole-image SpLiCE encodings. SpLiCE's inherently sparse concept space aids interpretability despite its large vocabulary. Our method (OCB) generates concepts from multiple object proposals, yielding a slightly less sparse concept space than the base CBM. To test if this increased capacity drives performance gains, we include a CBM (equal capacity) variant with reduced SpLiCE sparsity regularization to match OCB 's number of active concepts. We also compare to the Coarse-to-Fine CBM (C2F-CBM) (Panousis et al., 2024), which uses hierarchical, patch-based concepts. For fairness, we adopt the same LAION-based vocabulary as OCB and disable OCB sparsity constraints. Since C2F-CBM supports only single-label classification, it is evaluated

Table 1: **OCB shows improved task performance even for complex tasks.** OCB achieves superior performance compared to non-object-centric CBMs (based on SpLiCE) with both object-proposal generators, even compared to an equal concept capacity CBM. The best ("•") and runner-up ("○") results are bold.

| Model | Multi-label | | | Single-label | |
| --- | --- | --- | --- | --- | --- |
| | PASCAL-VOC | COCO(h) | COCO(l) | SUN397 | COCOLogic |
| CLIP+MLP | $89.60_{\pm 0.05}$ | $89.67_{\pm 0.03}$ | $68.20_{\pm 0.13}$ | $79.62_{\pm 0.11}$ | $65.95_{\pm 1.00}$ |
| CBM | $82.42_{\pm 0.01}$ | $84.73_{\pm 0.00}$ | $59.19_{\pm 0.00}$ | $74.79_{\pm 0.01}$ | $58.84_{\pm 0.09}$ |
| CBM (equal capacity) | $82.90_{\pm 0.01}$ | $86.31_{\pm 0.00}$ | $61.70_{\pm 0.00}$ | $74.85_{\pm 0.01}$ | $60.01_{\pm 0.09}$ |
| **OCB** (RCNN) | •$85.75_{\pm 0.01}$ | •$87.33_{\pm 0.00}$ | •$64.12_{\pm 0.00}$ | •$75.28_{\pm 0.04}$ | •$68.84_{\pm 0.11}$ |
| **OCB** (SAM) | ○$84.37_{\pm 0.01}$ | ○$86.91_{\pm 0.01}$ | ○$63.33_{\pm 0.00}$ | ○$75.13_{\pm 0.02}$ | ○$62.42_{\pm 0.10}$ |

on SUN397 and COCOLogic. Lastly, we compare to an opaque model that maps CLIP image embeddings directly to class predictions via a two-layer MLP (CLIP+MLP).

## 5.1 Enhancing Task Performance via Objects (RQ1).

To investigate OCB's general predictive performance, we compare the performance of CBMs with and without objects on the five different datasets. As the results in Tab. 1 show, the performance of OCB is superior to the non-object-centric CBM over all datasets, *i.e.*, both across the multi-label as well as single-label settings (we hereby focus on the OCB (RCNN) results and compare these to OCB (SAM) in the next subsection). Interestingly, OCB outperforms the base CBM even on SUN397, despite that dataset consisting of many classes like "sky", "desert sand", or "mountain", where an object-centric approach intuitively would not provide many benefits. However, the performance increase is rather small compared to the improvement on the more object-based tasks. The largest improvement can be seen on COCOLogic, which requires object-based visual reasoning. For COCO(l), we also observe a large boost via object representations, indicating that OCB successfully extends the concept space by information about specific objects in the image. To better contextualize our results, we also compare against a non-interpretable upper bound that uses the same image backbone as our models (CLIP+MLP). We observe that CLIP+MLP achieves higher accuracy, particularly than vanilla concept-based approaches, at the cost of offering no meaningful explanations for its predictions. However, our object-centric CBM (OCB) substantially narrows this gap: by leveraging structured object-level concept representations, OCB recovers much of the predictive power of the opaque model while retaining transparency.

As Tab. 1 indicates that more concept capacity improves the baseline performance, we compare base CBM and OCB with different concept capacities (i.e., the average number of non-zero concepts, cf. Fig. 8 in the appendix). Overall, increasing concept capacity improves the performance of the baseline and OCB, but on all object-based tasks, adding object-level concepts is consistently superior to adding more image-level concepts, again indicating that OCB allows for more competitive CBMs.

Lastly, in Tab. 2 we evaluate against the recent Coarse-to-Fine CBM (Panousis et al., 2024), which introduces hierarchical, patch-based image processing. As C2F does not support multi-label classification, we perform these evaluations on the SUN397 and COCOLogic datasets and, for fair comparisons, utilise a common LAION-based concept vocabulary with no sparsity constraints. We observe that our OCB model consistently outperforms C2F under these conditions, highlighting the benefit of object-centric representations over patch-based hierarchies.

Table 2: **Comparison between patch vs object encodings.** Results for SUN397 and COCOLogic datasets. Best results are in bold.

| Model | SUN397 | COCOLogic |
| --- | --- | --- |
| C2F-CBM | $70.12_{\pm 0.24}$ | $65.81_{\pm 1.57}$ |
| OCB (non-sparse) | $\mathbf{89.60}_{\pm 0.57}$ | $\mathbf{68.63}_{\pm 0.42}$ |

## 5.2 Impact of the Object-Proposal Model (RQ2).

As the object-proposal model is an important component of OCB, we compare two different instantiations of this component. Mask-RCNN (He et al., 2017) (RCNN) is pretrained on a rather narrow and specific domain, while Segment Anything (Kirillov et al., 2023) (SAM) can be considered a

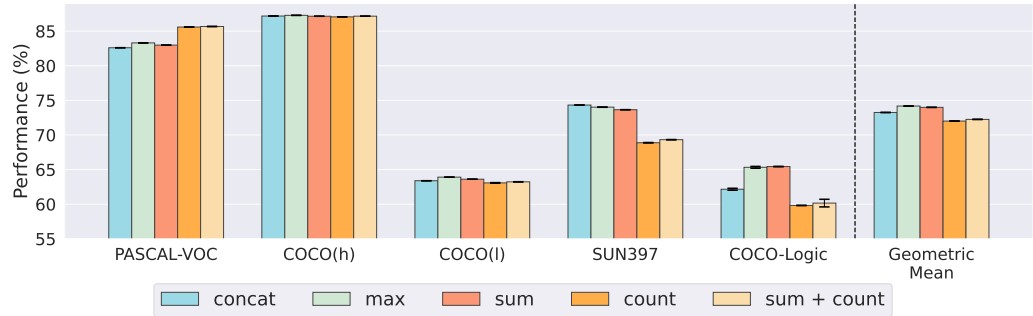

Figure 4: **Different datasets benefit from different aggregation methods.** While the performance comparison of different aggregation strategies with OCB (RCNN) and $k = 7$ does not show a one-fits-all choice, max or sum are always solid choices.

general-purpose object-proposal model. The results in Tab. 1 show that OCB outperforms the CBM baselines with either object detector. Additionally, we observe that the performance of in-domain object-proposal models surpasses the performance of the general-purpose object-proposal model. In particular, on COCOLogic, which requires the detection of only several specific object classes, the performance of OCB (RCNN) shines. Interestingly, the performance of OCB (RCNN) even surpasses OCB (SAM) on SUN397, which is mostly out-of-domain for Mask-RCNN. However, the difference between the two models is minimal, which can be explained by the general lack of object-centricity of SUN397. Overall, the results show that access to an in-domain object-proposal model can be beneficial for OCB, but using a general-purpose model instead also leads to good improvements.

### 5.3 Impact of Aggregation Mechanism (RQ3).

A second key element of OCB is the aggregation of the object-concepts and the image-level concepts. To investigate the impact of the different aggregation methods, we compare the task performance of OCB using Mask-RCNN as object encoder and a maximum number of objects $k = 7$. The results can be found in Fig. 4, with detailed numerical values in Sec. E. We observe that the concatenation of concept encodings generally does worse than sum or max. This is most likely due to the increased size of the concept space via concatenation (vocab size $\times$ num objects), and thus the input to the predictor network, making training the linear layer less effective. max and sum, on the other hand, perform quite similarly for all datasets. While max performs slightly better on the datasets that only require general information about the presence or absence of objects, sum has a slight advantage when object counts are also relevant. count and sum + count also perform very similarly, with the latter being consistently a bit better than the former, indicating that the additional continuous concept activations provide only a small benefit over the discrete counting information. On SUN397 and COCOLogic, the more discrete nature of count leads to overfitting of the predictor and, in turn, to a lower performance. On the other hand, for PASCAL-VOC, this property leads to a substantial increase in performance over the other baselines.

Although max aggregation leads to the best overall performance, the results indicate that there is no one-fits-all choice of the aggregation method. sum and max perform generally well, while sum + count can have the potential for good performance, but with the caveat of more difficult training of the predictor network. Additionally, it is noteworthy that concat retains full traceability between objects and concepts for model explanations, which might outweigh the smaller performance gain depending on the application (cf. discussion in Sec. 6).

### 5.4 Impact of the Object Proposal Refinement (RQ4).

Let us now turn to the influence of the number of object proposals for OCB. The most influential element of the refinement is the limitation of the number of objects to the top $k$. We investigate the impact of this parameter on the performance of OCB in Fig. 5, where the best-performing aggregation method is shown for each dataset. In general, we observe a trend indicating that adding more objects improves performance, in particular for PASCAL-VOC and COCO(h). On the other hand, performance beyond a certain point degrades again for COCOLogic and SUN397.

Due to the structure of COCOLogic, only the recognition of few key objects is necessary for a correct classification, while several classes require category counts or exclude certain categories. Thus, additional low-quality proposals, such as faulty or redundant detections, can lead to incorrect predictions. A similar effect occurs on the scene-centric SUN397, where extra object crops add noise rather than informative concepts. We also investigate the effect $t_{\min}$, the threshold limiting the minimum object size in Tab. 10 in the appendix. Here, it can be observed that a lower threshold generally increases performance, with the exception of SUN397, where the scene-centric nature of this dataset does not benefit from the detection of small objects. In summary, allowing more object proposals and the detection of smaller objects generally improves on object-centric datasets, careful filtering is essential to mitigate noise in tasks sensitive to false positives.

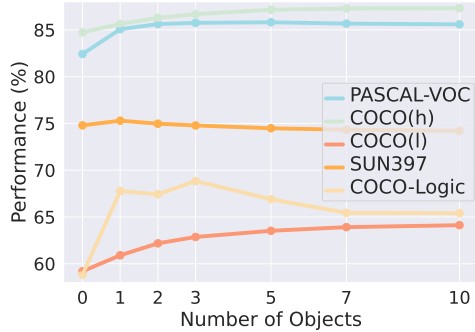

Figure 5: **Adding more object proposals improves performance on object-based tasks.** However, for complex tasks like COCOLogic, adding to many (noisy) object proposals can result in reduced performance. OCB (RCNN) performance for different values of $k$.

## 5.5 Interpreting Object-centric Concepts.

In the previous sections, we investigated whether the addition of object-centric concepts to CBMs can increase task performance. However, another important aspect of CBMs is their inherent interpretability via inspections of the concept space. Indeed OCB's concept space also allows for a more fine-grained and modular analysis of these concepts. As each object proposal is directly associated with the discovered concepts, it is possible to provide a conceptual description for each found object. Fig. 6 (top) shows an example where the object-centric concepts enrich the conceptual explanation of the image via previously undiscovered concepts, like the presence of a fork or a wine glass. In contrast, Fig. 6 (bottom) highlights that the object-centric concepts do not always have to extend the concept space by additional concepts. Even if the main concept (Pelican) is already identified in the full image encoding, the object-centric explanations reveal that there is not just one, but two pelicans in the image. Overall, the object-centric concept space allows for a more fine-grained representation of the image, also including information like object counts, which is an important basis for object-centric explanations.

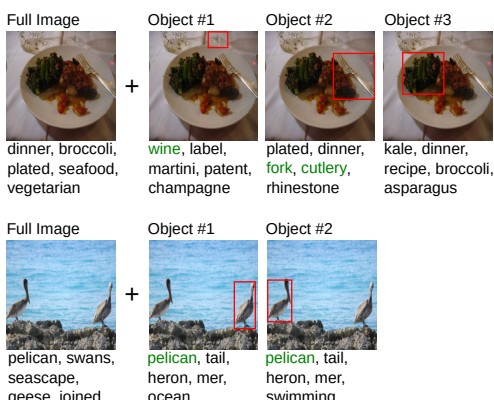

Figure 6: **Object-centric concept representations allow for a more fine-grained and modular concept space**. Object-centric representations include previously undiscovered concepts (top) or provide detailed information about object counts (bottom).

## 6 Discussion

Overall, we observe that integrating object representations into concept bottlenecks via OCB provides substantial performance improvements, but also explanatory value. However, there are a few points to take into consideration, which we wish to discuss here.

The aggregation step in OCB enables concept representations that are both fixed in size, regardless of the number of detected objects, and invariant to the order in which objects are detected. However, this comes at a potential cost: the ability to clearly map concepts back to individual objects may be lost. If two objects share overlapping concept representations, it becomes unclear which object a concept refers to. This limitation is especially pronounced when using aggregation methods like sum, max, count, and sum + count. In contrast, the concat aggregation preserves object-level associations

but suffers from a linearly growing encoding space with increasing number of objects. Overall, this highlights a trade-off between producing a compact concept representation and maintaining direct object identifiability; an important consideration in the context of object-level traceability.

To find suitable object proposals and extract meaningful concepts from them, OCB utilizes pretrained models to avoid annotations and additional training costs. While this is a big advantage, the performance of OCB is also dependent on the performance of these pretrained models. As shown in our experimental evaluations, the utilization of SAM as a general-purpose object proposal model is less performant compared to more specific models like RCNN. Similarly, the concept discovery module in our evaluations is based on SpLiCE, and thus on CLIP. While this approach can provide reasonable concept descriptions for a full image, cropping the image to an object proposal sometimes leads to noisy concepts, in particular if the object is small. While exploring other concept discovery models or other ways of encoding the image encodings with CLIP (*i.e.*, via pointing (Shtedritski et al., 2023)) could lead to a more robust concept discovery even for smaller objects, the overall performance remains bound by CLIP. In this context, some recent studies have raised concerns about the uncritical use of context concept-based models for interpretability. Debole et al. (2025) find that VLM-supervised CBMs can produce low-quality concepts with poor alignment to expert annotations, even when achieving strong downstream performance. Other works (Marconato et al., 2023; Bortolotti et al., 2025) raise concerns regarding shortcut learning in concept-based models. While OCB's more modular and object-centric concept space does not mitigate these limitations, it allows for more detailed control and evaluation of the learned concepts.

Lastly, we wish to highlight the importance of the global image encoding in overall task performance. As shown in an ablation study (Tab. 4), removing the image-level representation leads to a substantial drop in accuracy on Pascal VOC, despite the presence of rich object-centric features. These results reinforce a central message of our work: both object-centric and global features are necessary to achieve high predictive performance in complex visual reasoning tasks.

## 7 Conclusion

In this work, we show the importance of object-based encodings in the context of recent interpretable, concept-based models. The strength and novelty of our novel Object-Centric Concept Bottlenecks lie in making CBMs more structured, scalable, and interpretable by rethinking their core representation; from flat, image-level representations to grounded, object-level ones without requiring custom supervision or architectures. Our experimental evaluations show that this allows to solve more complex object-driven tasks such as multi-label classification and single-label object-level reasoning on our novel COCOLogic benchmark. Future avenues moving forward include investigating relational representations (Wüst et al., 2024; Delfosse et al., 2024), but also the potential of object-centric representations in other forms of interpretability research (Wäldchen et al., 2024), self-refinement (Stammer et al., 2024a) and more complex visual tasks such as visual question answering.

**Acknowledgments**

This work was supported by the "ML2MT" project from the Volkswagen Stiftung, the Priority Program (SPP) 2422 in the subproject "Optimization of active surface design of high-speed progressive tools using machine and deep learning algorithms" funded by the German Research Foundation (DFG) and supported by the DFG under Germany's Excellence Strategy (EXC 3066/1 "The Adaptive Mind", Project No. 533717223). It has further benefited from the HMWK projects "The Third Wave of Artificial Intelligence - 3AI", and Hessian.AI, the Hessian research priority program LOEWE within the project WhiteBox, the EU-funded "TANGO" project (EU Horizon 2023, GA No 57100431), and from early stages of the Cluster of Excellence "Reasonable AI" funded by the German Research Foundation (DFG) under Germany's Excellence Strategy— EXC-3057; funding will begin in 2026.

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

# Supplementary Materials


## A  Impact Statement

This paper introduces Object-Centric Concept Bottlenecks (OCB), a novel framework that advances the capabilities of concept-based models (CBMs) by incorporating object-centric representations. By leveraging pretrained object detection and concept discovery modules, OCB enables interpretable, structured, and high-performing decision-making on complex visual tasks, such as multi-label and logic-based image classification, without requiring extensive manual annotations. Notably, it extends the applicability of CBMs beyond traditional single-label settings, improving both predictive accuracy and explanation granularity.

Ethically, the approach supports greater transparency and control in AI systems by producing explanations tied to discrete objects and their associated concepts. This aligns with responsible AI principles, especially in safety-critical applications such as autonomous systems. However, OCB also inherits limitations from the pretrained models it builds upon. These models may encode biased or misaligned representations (e.g., due to skewed training data in foundation models like CLIP or SAM), potentially resulting in misleading or culturally insensitive explanations. The framework's reliance on such components emphasizes the need for careful auditing and user-in-the-loop validation in real-world deployments.

In summary, OCB presents an important step toward more interpretable and modular visual reasoning systems, while raising critical questions about the trustworthiness and accountability of explanations generated by models grounded in potentially flawed foundation systems.

## B  Additional Details for OCB

Alg. 1 outlines the procedure used by OCB to refine the object proposals generated by an object-proposal model. These proposals consist of pairs of bounding boxes and certainty scores. They are first filtered by bounding box size and a certainty threshold to discard uninformative or low-confidence candidates. The remaining proposals are sorted by certainty score and filtered based on intersection-over-union (IOU), removing boxes that significantly overlap with higher-scoring proposals. Finally, the number of proposals is limited to a predefined maximum $k$.

---
**Algorithm 1** Refine Object Proposals

---
**Require:** Image $x$, object-proposal model $o$, thresholds $t_{\min}, t_{\max}, t_{cer}, t_{IOU}$, max proposals $k$
1:   $(B, S) \leftarrow o(x)$
2:   $F \leftarrow [\,]$                                                    ▷ filter by size and certainty
3:   **for all** $(b_i, s_i) \in (B, S)$ **do**
4:       **if** $t_{\min} \leq \text{size}(b_i) \leq t_{\max}$ **and** $s_i \geq t_{cer}$ **then**
5:          Append $(b_i, s_i)$ to $F$
6:       **end if**
7:   **end for**
8:   Sort $F$ in descending order of score $s$
9:   $N \leftarrow [\,]$                                                   ▷ remove overlapping boxes
10: **for all** $(b_i, s_i) \in F$ **do**
11:     **if** for all $b_j \in N$, $\text{IoU}(b_i, b_j) \leq t_{\text{IoU}}$ **then**
12:       Append $b_i$ to $N$
13:     **end if**
14: **end for**
15: $B_r \leftarrow (n_1, \cdots, n_k)$                                 ▷ first $k$ elements of N
16: **return** $B_r$

---

## C  Additional Details for Evaluations

In our evaluations, we compare OCB against a baseline CBM and evaluated the performance of both on several different datasets. As concept extractor for OCB and the baseline CBM we use SpLiCE (Bhalla et al., 2024). We followed their default settings, for model, vocabulary and l1-regularization. The only exception is to that is CBM (equal capacity), where we reduced the regularization from 0.25 to 0.2, whcih led to a comparable number of non-zero concepts to OCB.

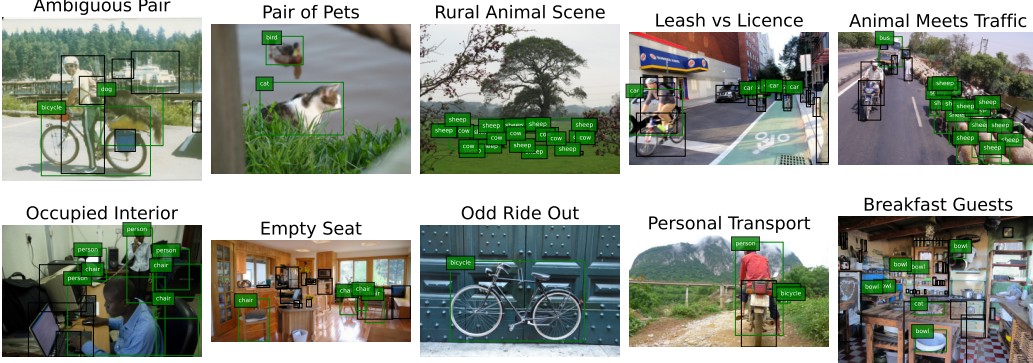

Figure 7: Example images of the ten classes from COCOLogic. The relevant objects for the classes are marked green.

OCB also introduces several hyperparameters: While we provide ablations on the number of objects-porposals $k$ and the choice of aggregation, we kept the hyperparameters $t_{min}$, $t_{max}$ and $t_{cer}$ fixed. For OCB (RCNN), $t_{min} = 0.01$, $t_{max} = 0.85$ and $t_{cer} = 0.2$, where the size-related parameters consider the bounding box size relativ to the full image size. For OCB (SAM), $t_{min} = 0.02$, $t_{max} = 0.85$ and $t_{cer} = 0.94$, and we used the stability factor of SAM as certainty score. Additionally, we used a grid-like prompting scheme for SAM with 16 points per iamge to generate object-proposales. $t_{IoU} = 0.5$ is kept the same for both object proposal models.

For the predictor network, we used a single linear layer. For its training, we optimized the parameters learning rate and the number of epochs the layer was trained for, using the validation sets. The parameter configurations for each dataset and aggregation can be found in the code repository. All experiments were conducted using T single GPUs from Nvidia DGX2 machines equipped with A100-40G and A100-80G graphics processing units.

## D COCOLogic

The COCOLogic dataset comprises ten semantically rich classes derived from COCO images. Examples of each class are shown in Fig. 7. Each class is defined by a specific logical rule, detailed in Tab. 3 alongside the number of training and test examples per class. Images are selected such that exactly one class applies per image; any image that does not satisfy exactly one rule is discarded. This ensures that the class labels are mutually exclusive and unambiguous.

These rules are designed to be *semantically meaningful* while introducing *logical complexity* that makes the classification task *non-linearly separable* in the input space. Consequently, COCOLogic serves as a challenging benchmark for assessing the representational capacity of both linear and non-linear classifiers, as well as symbolic and neuro-symbolic models. In this work we mainly evaluated models that had linear classifiers (both OCB and the baseline CBM) on COCOLogic. While this showed that incorporating object-level concepts are essential to solve this dataset, utilizing more powerful classifiers is equally as important, as linear classifiers cannot resolve the more complex concept-class relationships.

## E Additional Evaluations

In Tab. 4, we ablate the influence of the whole image encoding versus object encodings in OCB. We observe that removing the image-level encodings from OCB leads to a substantial performance drop (Pascal VOC), confirming that object-centric and global representations provide complementary information. These results support a key message of our work: both object-centric and global features are necessary for achieving high predictive performance.

Table 3: COCOLogic class definitions and sample sizes.

| Class Name | Class Rule | Samples |
|---|---|---|
| Ambiguous Pairs | (cat XOR dog) AND (bicycle XOR motorcycle) | 36 |
| Pair of Pets | Exactly two categories of {cat, dog, bird} are present | 56 |
| Rural Animal Scene | At least one of {cow, horse, sheep} AND no person | 2965 |
| Leash vs Licence | dog XOR car | 4188 |
| Animal Meets Traffic | At least one of {horse, cow, sheep} AND at least one of {car, bus, traffic light} | 24 |
| Occupied Interior | (couch OR chair) AND at least one person | 8252 |
| Empty Seat | (couch OR chair) AND no person | 4954 |
| Odd Ride Out | Exactly one category of {bicycle, motorcycle, car, bus} | 3570 |
| Personal Transport | person AND (bicycle XOR car) | 279 |
| Breakfast Guests | bowl AND at least one of {dog, cat, horse, cow, sheep} | 169 |

Table 4: Performance comparison on Pascal VOC showing the contribution of object-centric and image-level encodings.

| | Full (Object + Image) | Only Image Encodings | Only Object Encodings |
|---|---|---|---|
| VOC | $85.75 \pm 0.01$ | $82.42 \pm 0.01$ | $77.48 \pm 0.03$ |

### E.1 Analysis of CBM Capacities

As the evaluation in table Tab. 1 shows, increasing the capacity of a CBM generally also improves its performance. In that, we refer to the capacity as the average number of non-zero concepts in its concept space. In Fig. 8, we conduct a more thorough evaluation of the effect of concept capacity on model performance. The results confirm that adding more concept capacity generally also improves performance, with the exception of COCOLogic, where there is a dropoff after a certain concept capacity. While this trend is true both for the base CBM and for OCB, adding more concept capacity via object-level concepts is better on object-based task than just increasing the number of image-level concepts and hope that the objects are going to be represented better. For COCOLogic, both OCB and the base CBM have a dropoff in performance after a certain number of concepts in the concept space, which could be explained by the additional noisy concepts making this already challenging task more difficult and do not provide benefits anymore.

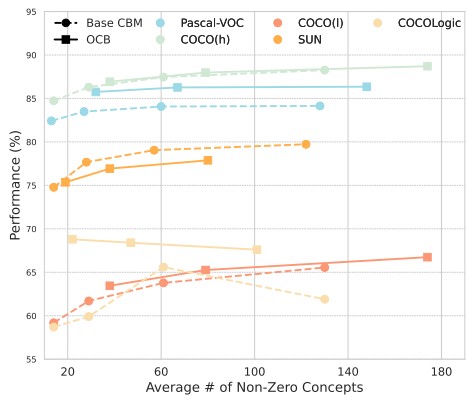

Figure 8: **Adding object-level concepts is generally better than increasing image-level concepts.** Comparison of the base CBM and OCB with different levels of sparsity regularization (resulting in a different number of non-zero concepts). For all object-based tasks, adding object-level results in bigger performance gains than just adding more image-level concepts.

As both models use SpLiCE as a backbone, the exact number of non-zero concepts cannot directly be set. Instead, SpLiCE uses l1-regularization to enforce the sparsity of the concept space. Setting the l1-regularization strength does result in different average non-zero concepts which is the reason for the differences in Fig. 8.

Table 5: VOC Results OCB (RCNN)

| Num Objects | 1 | 2 | 3 | 5 | 7 | 10 |
|---|---|---|---|---|---|---|
| Sum | $84.15 \pm 0.01$ | $83.87 \pm 0.01$ | $83.28 \pm 0.01$ | $82.84 \pm 0.00$ | $82.98 \pm 0.01$ | $83.11 \pm 0.00$ |
| Max | $84.28 \pm 0.01$ | $84.22 \pm 0.01$ | $83.77 \pm 0.00$ | $83.26 \pm 0.01$ | $83.30 \pm 0.01$ | $83.34 \pm 0.01$ |
| Concat | $84.04 \pm 0.01$ | $83.90 \pm 0.01$ | $83.27 \pm 0.01$ | $82.75 \pm 0.01$ | $82.60 \pm 0.01$ | $82.41 \pm 0.01$ |
| Sum Count | $85.07 \pm 0.00$ | $85.62 \pm 0.00$ | $85.75 \pm 0.01$ | $85.81 \pm 0.01$ | $85.67 \pm 0.01$ | $85.59 \pm 0.01$ |
| Count | $84.96 \pm 0.00$ | $85.52 \pm 0.00$ | $85.66 \pm 0.00$ | $85.72 \pm 0.01$ | $85.59 \pm 0.01$ | $85.51 \pm 0.01$ |

Table 6: COCO(l) results OCB (RCNN)

| Num Objects | 1 | 2 | 3 | 5 | 7 | 10 |
|---|---|---|---|---|---|---|
| Sum | $60.78 \pm 0.00$ | $62.19 \pm 0.01$ | $62.85 \pm 0.00$ | $63.42 \pm 0.00$ | $63.61 \pm 0.00$ | $63.82 \pm 0.01$ |
| Max | $60.90 \pm 0.00$ | $62.18 \pm 0.03$ | $62.86 \pm 0.01$ | $63.52 \pm 0.00$ | $63.91 \pm 0.00$ | $64.12 \pm 0.00$ |
| Concat | $61.14 \pm 0.00$ | $62.24 \pm 0.00$ | $62.82 \pm 0.00$ | $63.22 \pm 0.01$ | $63.39 \pm 0.00$ | $63.43 \pm 0.00$ |
| Sum + Count | $61.50 \pm 0.00$ | $62.44 \pm 0.03$ | $62.77 \pm 0.01$ | $63.15 \pm 0.00$ | $63.23 \pm 0.02$ | $63.29 \pm 0.03$ |
| Count | $61.30 \pm 0.00$ | $62.28 \pm 0.00$ | $62.57 \pm 0.01$ | $62.95 \pm 0.01$ | $63.08 \pm 0.04$ | $63.12 \pm 0.02$ |

Table 7: COCO(h) results, OCB (RCNN)

| Num Objects | 1 | 2 | 3 | 5 | 7 | 10 |
|---|---|---|---|---|---|---|
| Sum | $85.57 \pm 0.00$ | $86.26 \pm 0.00$ | $86.65 \pm 0.00$ | $87.06 \pm 0.00$ | $87.15 \pm 0.00$ | $87.18 \pm 0.00$ |
| Max | $85.64 \pm 0.00$ | $86.28 \pm 0.00$ | $86.68 \pm 0.00$ | $87.15 \pm 0.00$ | $87.28 \pm 0.00$ | $87.33 \pm 0.00$ |
| Concat | $85.82 \pm 0.00$ | $86.45 \pm 0.00$ | $86.79 \pm 0.00$ | $87.08 \pm 0.00$ | $87.18 \pm 0.00$ | $87.19 \pm 0.00$ |
| Sum + Count | $86.01 \pm 0.00$ | $86.55 \pm 0.00$ | $86.91 \pm 0.00$ | $87.17 \pm 0.00$ | $87.16 \pm 0.00$ | $87.11 \pm 0.00$ |
| Count | $85.89 \pm 0.00$ | $86.45 \pm 0.00$ | $86.81 \pm 0.00$ | $87.07 \pm 0.00$ | $87.06 \pm 0.00$ | $87.01 \pm 0.00$ |

Table 8: COCO-Logic results, OCB (RCNN)

| Num Objects | 1 | 2 | 3 | 5 | 7 | 10 |
|---|---|---|---|---|---|---|
| Sum | $67.77 \pm 0.08$ | $67.42 \pm 0.11$ | $68.84 \pm 0.11$ | $66.89 \pm 0.08$ | $65.44 \pm 0.03$ | $65.40 \pm 0.12$ |
| Max | $64.31 \pm 0.99$ | $66.72 \pm 0.09$ | $68.67 \pm 0.13$ | $66.97 \pm 0.15$ | $65.33 \pm 0.13$ | $65.23 \pm 0.06$ |
| Concat | $65.27 \pm 0.13$ | $65.67 \pm 0.08$ | $66.32 \pm 0.56$ | $63.07 \pm 0.04$ | $62.16 \pm 0.14$ | $60.18 \pm 0.21$ |
| Sum + Count | $67.37 \pm 0.07$ | $63.77 \pm 0.06$ | $61.83 \pm 0.12$ | $61.56 \pm 0.09$ | $60.16 \pm 0.55$ | $61.38 \pm 0.78$ |
| Count | $67.42 \pm 0.13$ | $63.85 \pm 0.08$ | $61.97 \pm 0.10$ | $61.57 \pm 0.11$ | $59.82 \pm 0.02$ | $61.34 \pm 0.89$ |

Table 9: **No one-fits-all choice of the aggregation method.** Comparing the performance of different aggregation strategies with OCB (RCNN) and $k = 7$.

| Aggregation | PASCAL-VOC | COCO(h) | COCO(l) | SUN397 | COCO-Logic | Geometric Mean |
|---|---|---|---|---|---|---|
| concat | $82.60_{\pm 0.01}$ | $87.18_{\pm 0.00}$ | $63.39_{\pm 0.00}$ | $\mathbf{74.32}_{\pm 0.01}$ | $62.16_{\pm 0.14}$ | $73.25$ |
| max | $83.30_{\pm 0.01}$ | $\mathbf{87.28}_{\pm 0.00}$ | $\mathbf{63.91}_{\pm 0.00}$ | $74.02_{\pm 0.02}$ | $65.33_{\pm 0.13}$ | $\mathbf{74.18}$ |
| sum | $82.98_{\pm 0.01}$ | $87.15_{\pm 0.00}$ | $63.61_{\pm 0.00}$ | $73.65_{\pm 0.01}$ | $\mathbf{65.44}_{\pm 0.03}$ | $73.99$ |
| count | $85.59_{\pm 0.01}$ | $87.06_{\pm 0.00}$ | $63.08_{\pm 0.04}$ | $68.88_{\pm 0.03}$ | $59.82_{\pm 0.02}$ | $72.01$ |
| sum + count | $\mathbf{85.67}_{\pm 0.01}$ | $87.16_{\pm 0.00}$ | $63.23_{\pm 0.02}$ | $69.32_{\pm 0.02}$ | $60.16_{\pm 0.55}$ | $72.25$ |

## E.2 Object Number and Aggregation

In this section we provide further results for varying object numbers and aggregation strategies. In Tab. 9, the numerical values for Fig. 4 are reported. Additionally, we have results forPASCAL-VOC in Tab. 5, COCO(l) in Tab. 6, COCO(h) in Tab. 7 and COCOLogic in Tab. 8 for Mask RCNN as object proposal method.

## E.3 Minimum Object Size

Next to the aggregation method and the $k$ (i.e. the maximum number of potential objects allowed), we investigate here the impact of the parameter $t_{\min}$ on the performance of OCB. This threshold sets the minimum size of an object proposal relative to the image size. In Tab. 10, we show the performance of OCB on the different datasets when varying the minimum size of an object proposal. Overall, one can see that lowering this minimum size consistently improves performance. The only exception is SUN, where there is a slight improvement with higher minimum object size. This

| $t_{\min}$ | Multi-label | | | Single-label | |
|---|---|---|---|---|---|
| | PASCAL-VOC | COCO(h) | COCO(l) | SUN397 | COCOLogic |
| Baseline | $82.42_{\pm 0.01}$ | $84.73_{\pm 0.00}$ | $59.19_{\pm 0.00}$ | $74.79_{\pm 0.01}$ | $58.84_{\pm 0.09}$ |
| 0.005 | $85.87_{\pm 0.17}$ | $87.53_{\pm 0.00}$ | $64.34_{\pm 0.00}$ | $75.24_{\pm 0.02}$ | $69.31_{\pm 0.09}$ |
| 0.01 | $85.70_{\pm 0.11}$ | $87.32_{\pm 0.01}$ | $64.11_{\pm 0.00}$ | $75.29_{\pm 0.03}$ | $68.76_{\pm 0.07}$ |
| 0.02 | $85.21_{\pm 0.12}$ | $86.93_{\pm 0.00}$ | $63.45_{\pm 0.01}$ | $75.35_{\pm 0.02}$ | $66.20_{\pm 0.04}$ |
| 0.05 | $84.62_{\pm 0.04}$ | $86.40_{\pm 0.00}$ | $62.32_{\pm 0.00}$ | $75.61_{\pm 0.02}$ | $67.70_{\pm 0.69}$ |

Table 10: Investigating the effect of the minimum object size threshold ($t_{\min}$) on the performance of OCB. All runs are over 5 seeds and with Mask-RCNN and the best performing settings of aggregation and $k$.

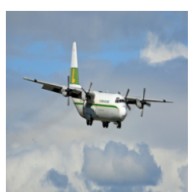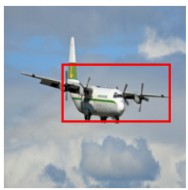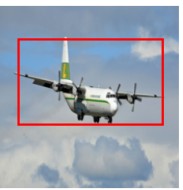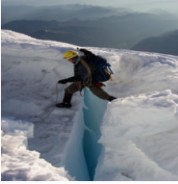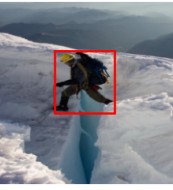

Figure 9: Typical failure cases of the object-proposal generation of OCB. (Left): Multiple object proposals of the same object get generated and are not filtered out. (Right): Irrelevant objects for the context of the task get detected and result in spurious associations, i.e., a human gets associated with the class "snow field" of SUN.

can be explained by the task of SUN - scene understanding often does not need small objects, so increasing the minimum object size removes non-important object proposals. While lower values of $t_{\min}$ are generally beneficial, it thus remains important to consider the data and task at hand and decide whether detecting smaller objects is helpful.

### E.4 Typical Failure Cases of the Object Proposals

While the pre-trained object-proposal generation of OCB allows for quite a general detection of potential objects without costly training, there are also some typical failure cases of these detectors.

Sometimes, it can happen that the object proposal model generates multiple proposals from the same object, despite filtering them for a low overlapping IOU. In this case, the same object appears in the concept space multiple times, which can lead to incorrect results if the exact number of concepts is relevant (cf. Fig. 9 (left), where multiple proposals of the airplane are generated and not filtered).

If some objects are inherently not relevant for the task, detecting them and adding them to the concept space can introduce spurious correlations. For instance, in the "snow field" class of the SUN dataset, most samples also contain humans, which are detected as objects (cf. Fig. 9, right). This can lead the model to incorrectly associate the presence of humans with the "snow field" label.

