# OpenReview forum: "Object-Centric Concept-Bottlenecks"
_NeurIPS.cc/2025/Conference — NeurIPS 2025 poster_

### Official Review · Reviewer_Kyb9 · 2025-07-02

**Clarity:** 2
**Significance:** 1
**Originality:** 3
**Rating:** 4
**Confidence:** 3

**Summary:**

The work developed a new framework (Object-Centric Concept Bottlenecks--OCB) for visual reasoning tasks, using "object-level representations".
It identifies objects in an image and then infers concepts from the image and the identified objects before making the final prediction.

**Questions:**

> Fig. 6 (top) shows an example where the object-centric concepts enrich the conceptual explanation of the image via previously undiscovered concepts, like the presence of a fork or a wine glass.
- Could this not be viewed as that CBM does not consider them as relevant concepts or as less important concepts?

- Why use the same concept module for the whole image and the resized-cropped object region? Conceptually, if object-level and image-level are supposed to be different granularity of concepts, does it not make sense that the concepts extracted should be from different concept sets?

> Interestingly, OCB outperforms the base CBM even on SUN397, despite that dataset consisting of many classes like "sky", "desert sand", or "mountain", where an object-centric approach intuitively would not provide many benefits."
- What do the concepts look like in these classes? More generally, how does "object-level concepts" work for relations, actions, properties etc, that are not object based, in terms of accuracy and the level of interoperability?

- Why stop at two levels? Why not keep going down the object hierarchy to get more specific breakdowns of objects?

> While OCB’s more modular and object-centric concept space does not mitigate these limitations, it allows for more detailed control and evaluation of the learned concepts.
What kinds of control and evaluation?

Minor comments:

Line 112: citations seem mis-formatted

Line 195: missing parenthesis

**Ethical Concerns:**

["NO or VERY MINOR ethics concerns only"]

**Final Justification:**

The additional results are promising, and the authors addressed my main concerns about evaluation and the claims. The only drawback  is the limited application, where OCB only significantly benefit multi-objects-centric classification tasks.

**Limitations:**

yes

**Quality:**

1

**Strengths And Weaknesses:**

Strengths:
- Significance: The work presents a new benchmark for evaluating single-label classification requiring visual reasoning. OCB performs better than CBM on image classification tasks that require visual reasoning.

Weaknesses:
- Clarity: It is not clear how fundamentally object-level concepts are different from image-level concepts beyond that there's some vague notion of level of details. Object-level concepts seem just as possible to be incorporated into the concept space of image-level concepts. It seems like the novelty is having layers of concept detection, with some arbitrary hierarchical grouping, but this is not made explicit or clear.
- Significance: While OCB could be useful for classification tasks that rely on labels built on top of multiple objects, it does not seem to generalize to other classification tasks, as shown by the minor improvement over SUN, that do not segment nicely into objects with clear boundaries. It also does not perform significantly better over CBM (equal capacity) on tasks that do not need reasoning over multiple objects.
- Quality: The evaluation is limited. The work does not evaluate against common benchmarks such as ImageNet or CIFAR-100 nor many  other classification tasks in non-object domains aside from SUN.

---

> ### Author Rebuttal · Authors · 2025-07-31
>
> We thank the reviewer for their time and suggestions. We provide clarifications for these points below.
>
> > Clarification object-level vs image-level concepts in CBMs
>
> Traditionally, CBMs compute a single concept representation for the entire image, often relying on supervised concept labels [1] or CLIP-based methods that match global image features to textual descriptions [e.g., [2]]. The image-level concepts reflect properties of the overall scene but do not explicitly distinguish between individual entities within.
>
> In contrast, object-level concepts, as introduced in OCB, are computed separately for each detected object in the image. While both are based on the same backbone encoder, object-level concepts are grounded in **specific, localized entities**, enabling a structured representation of the scene. This difference is not merely about granularity; it provides **explicit grouping of concepts by object**, which is not available in global image-level approaches.
>
> This structure has important implications:
> - It enables reasoning over **multiple entities** in complex, multi-object scenes.
> - It allows models to distinguish **which concepts belong together**, tied to specific objects.
> - It facilitates **compositional and interpretable reasoning**, aligning more closely with how humans understand scenes.
>
> Our results support this distinction. For example, our **sparsity experiments** show that object-centric representations encode concepts more efficiently and faithfully than image-level representations. Moreover, object-centric models perform better in **multi-label settings**, where distinct object-level reasoning is essential.
>
> While object-level concepts could be forced into an image-level framework, this ignores the inductive bias that gives object-centric representations their strength. Our contribution goes beyond hierarchical grouping, it offers a practical, interpretable path from flat concept reasoning to **structured, entity-aware representations**, essential for scaling CBMs to real-world vision tasks.
>
> We have updated the manuscript to make sure that these points are presented more clearly in the paper.
>
> > 2. What is the novelty of this work?
>
> We acknowledge that the individual components of our Object-Centric Bottleneck (OCB) framework, such as object proposal generation and concept classification, are not novel in isolation. However, the novelty of our work does not lie in proposing a new object detection mechanism, but in how we integrate object-centric representations into the Concept Bottleneck Model (CBM) framework, making it scalable and applicable to complex, real-world visual data.
>
> While this design may appear simple, bridging object-centric representations with CBMs in a real-world setting is non-trivial and has not been demonstrated before. Prior work exploring object-centric inductive biases has largely been limited to synthetic and single-class environments. Our contribution lies in extending these insights to realistic, diverse, and noisy data, removing the need for manual concept annotations, improving cross-dataset generalization, and demonstrating that object-level concept reasoning is viable and beneficial in practical scenarios.
>
> In summary, the strength and novelty of our approach lie in making CBMs more structured, scalable, and interpretable by rethinking their core representation from flat, image-level concepts to grounded, object-level ones without requiring custom supervision or architectures.
>
>
> > Small improvements on tasks that do not require object information.
>
> The primary goal of OCB is to **extend the applicability of CBMs to more complex visual tasks**, such as multi-label classification or visual reasoning, where **reasoning over distinct objects is essential**.
> We had included results on SUN, which is commonly used in CBM evaluations, to demonstrate that **adding object-centric representations does not harm performance** on tasks where object structure is less relevant.
>
>
> > Evaluation against common benchmarks in non-object domains.
>
> As discussed in the previous point, the goal of OCB is not to improve performance on non-object-centric tasks, but to **enable concept-based models to operate effectively on object-centric, multi-entity domains**, a setting where traditional CBMs struggle.
> For this reason, our evaluation primarily focuses on datasets with explicit object structure, such as Pascal VOC and COCO. That said, we agree that broader benchmarking is valuable and present additional experiments on CIFAR-100, to expand the evaluation beyond object-segmented scenes. We observe that OCB and CBM perform comparably, indicating that OCB does not incur any performance loss on non-object-centric benchmarks. We have added the results to the paper.
>
> |CIFAR100|CBM|OCB|
> |-|-|-|
> |Acc|$74.21 \pm 0.01$|$74.47\pm 0.02$|
>
> ### Questions:
>
>
> > Re: Qualitative concept examples:
> 1. Does CBM view the object-concepts as less important?
> A: Yes, that is a valid interpretation. The base (image-level) CBM lacks the spatial focus to attend to smaller or less prominent objects like the fork or wine glass. As a result, these concepts may not be activated or may be deemed less relevant, even if they are present and potentially informative. In contrast, OCB explicitly examines individual regions, allowing it to **surface concepts tied to localized entities** that might be overlooked at the global level.
> 2. Why same concept module/vocabulary?
> A: We used a shared, general-purpose concept vocabulary to keep the approach task-agnostic and free of additional supervision. This ensures broad applicability without requiring domain-specific concept sets.
> That said, we agree that using different vocabularies for image-level and object-level representations when task-specific information is available is a promising direction for future work.
>
> > Re: Examples for concepts for SUN
>
> 1. Example concepts for SUN:
> For SUN, many images from natural scenes do not have any recognized objects. However, there are also scenes like "computer room", "stable" or "dining room" which have detectable objects like computers, horses, furniture or food that help identifying the global scene category. This explains the slight performance increase over the base model for this dataset.
>
> 2. Relations, Actions, Hierarchies?
> A: We see object-level representations as a foundational step toward more structured reasoning, including relations, actions, and hierarchical object decomposition. Our goal in this work was to establish object-centric concept representations within CBMs.
> While we stop at the object level, we agree that further decomposition into sub-parts, relationships, or action-level concepts which often **emerge from interactions between objects or between objects and the global scene** is a natural and valuable next step. Our current design provides a flexible base for such extensions, and we view this as a promising direction for future work.
>
> > Re: What kind of control and evaluation for concepts?
>
>  A: OCB enables more fine-grained control and evaluation by explicitly structuring the concept space around individual objects. In particular, when using the `concat` aggregation, the model retains which concepts are tied to which object proposals. This allows for **object-level explanations**, e.g., identifying which specific object triggered a certain prediction, and makes it possible to **evaluate concept activation per object**, rather than just at the global image level. This structured representation also allows for **targeted interventions**, such as modifying or masking concepts for a specific object.
>
> > Additional baseline results
>
> We additionally wish to highlight novel baseline and ablation evaluations resulting from other reviews and strengthen the findings of our work.
>
> ### 1. Opaque Upper-Bound Model
> We introduce a non-interpretable MLP classifier on top of frozen CLIP embeddings as an upper bound. As expected, it outperforms interpretable CBMs but lacks transparency. OCB narrows the gap significantly, offering strong accuracy with interpretable object-centric reasoning.
>
> | |Pascal-VOC|COCO(h)|COCO(l)|SUN|COCOLogic|
> |-|-|-|-|-|-|
> |CBM|$82.42\pm 0.01$|$84.73\pm 0.00$|$59.19\pm 0.00$|$74.79\pm 0.01$|$58.84\pm 0.09$|
> |**OCB (RCNN)**|$85.75\pm 0.01$|$87.33\pm 0.00$|$64.12\pm 0.00$|$75.28\pm 0.04$|$68.84\pm 0.11$|
> |Upper Bound|$89.60\pm 0.05$|$89.67\pm 0.03$|$68.20\pm 0.13$|$79.62\pm 0.11$|$65.95\pm 1.00$|
>
> ### 2. Coarse-to-Fine CBM (C2F)
> We compare OCB to Coarse-to-Fine CBM [4], using the same LAION concept vocabulary. Evaluated on SUN and COCOLogic (single-label tasks), OCB outperforms C2F, highlighting the benefits of object-centric over patch-based representations.
>
> | |SUN|COCOLogic|
> |-|-|-|
> |C2F|$61.28\pm 1.44$|$66.27\pm 1.09$|
> |OCB|$78.11\pm 0.57$|$68.63\pm 0.42$|
>
> ### 3. Sparsity Analysis
> We assess both CBM and OCB under varying sparsity constraints (via $\ell_1$ penalty). OCB consistently outperforms CBM at all sparsity levels, confirming that its gains stem from object-centric structure, not capacity.
>
> |# Non-Zero Concepts|15|25–30|60–70|130–160|No Sparsity|
> |-|-|-|-|-|-|
> |CBM|$82.42$|$83.49$|$84.07$|$84.14$|$88.27$|
> |OCB|–|$85.75$|$86.27$|$86.35$|$90.56$|
>
> We trust that these clarifications and additional results strengthen the clarity and completeness of our work, and we respectfully encourage the reviewer to reconsider their evaluation in light of these updates.
>
>
> [1] Koh et al. "Concept bottleneck models." ICML, 2020.
>
> [2] Yang et al. "Language in a bottle: Language model guided concept bottlenecks for interpretable image classification." CVPR, 2023.
>
> [3] Marconato et al. "Not all neuro-symbolic concepts are created equal: Analysis and mitigation of reasoning shortcuts." NeurIPS, 2023.
>
> [4] Panousis et al. "Coarse-to-Fine Concept Bottleneck Models." NeurIPS, 2024.

---

> > ### Comment · Reviewer_Kyb9 · 2025-08-04
> >
> > I thank the authors for the response. I have a few follow-up questions:
> >
> > Could the authors explain how OCB provides “fine-grained and modular concept space” or “high-level concepts” when the object-level and image-level concept spaces are the same?
> >
> > Concerning the claim about concept groupings,
> > > This difference is not merely about granularity; it provides explicit grouping of concepts by object, which is not available in global image-level approaches.
> >
> > > It allows models to distinguish which concepts belong together, tied to specific objects.
> >
> > However, the concept groupings by object boundaries are not actually employed during the prediction. The concepts are aggregated but the groupings are not maintained (except for `concat`, but even there the groupings are not conveyed explicitly if I am understanding correctly). Could the authors clarify how OCB enables the models to identify groupings?
> >
> > > It facilitates compositional and interpretable reasoning, aligning more closely with how humans understand scenes.
> >
> > What is the compositionality that OCB enables? It does not seem like different concept spaces can be easily composed together?
> >
> > > This allows for object-level explanations, e.g., identifying which specific object triggered a certain prediction, and makes it possible to evaluate concept activation per object, rather than just at the global image level.
> >
> > How can this be possible if the concepts from different objects end up aggregated using non-invertible functions? Could the authors give an example of these usages?

---

> > > ### Author Response · Authors · 2025-08-06
> > >
> > > We thank the reviewer for the follow-up questions, which helped to better understand the original concerns. Initially, we interpreted the questions as addressing the general benefits of object-centric representations. In light of your clarification, we now see that the reviewer was referring more specifically to the architectural details of OCB. As a result, our previous response may have been too general in scope.
> > >
> > > > How does OCB provide a "fine-grained and modular concept space" and "high-level concepts" when the object spaces are the same.
> > >
> > > - While the vocabulary for concepts is the same for image-level and object-level concepts, there is a difference in the way the concepts are generated:
> > > - The image-level concepts are generated via encoding the whole image with CLIP, and enforcing sparsity in the resulting concept representation: This effectively results in the image-level concepts containing the most prevalent elements of the image and generally do not contain much details.
> > > - While the object-level concepts are based on the same vocabulary, the image input is different. As only the image part containing the object is presented to CLIP, the object-level concepts describe elements of the given objects.
> > > - This effectively provides more detailed concepts, as the object-level concepts generally describe the object and not the whole image.
> > > - The modular aspect refers to the ability to add as many object encodings on the fly (concat excluded) as required.
> > >
> > > > Grouping & Compositional reasoning:
> > > - We had misinterpreted the initial questions. In general, OCB's concept space allows to trace concepts to objects which in turn enables forms of compositional reasoning. We agree that with the current setup, the predictor limits the level of complex reasoning over objects, due to the use of a linear prediction layer in the context of CBMs.
> > > - However, the more structured concept space of OCB (in particular when using concat as aggregation) allows to use predictors which rely on reasoning not just on a whole image level, but on an object-level as well. Currently, this setup allows to answer questions like "Are there more than five sheep in an image?". This would not be possible with only image-level encodings. Now, by replacing the linear layer, e.g., with a logical reasoner this would enable us to even answer questions like "There are three white and two black sheep.".
> > >
> > > > Object-level explanations:
> > > - While we agree that tracing the individual objects back to a model's prediction with other aggregation approaches than concat is more tricky (as we had noted in the paper), in principle methods like gradient-based importance mapping from classical XAI literature could provide support in these cases.
> > > - In general, when the concepts from objects do not completely overlap (which is often the case given the sparse concept activations), it is still possible to link attributions back to objects when using the other aggregations, as the activation of the concept comes from an unique object. E.g., based on the example in the top of Fig. 6 in the main paper if the class prediction is "Wine dinner" one part of the explanation could be "because of wine" which can be traced back to Object \#1.
> > >
> > > We hope our response can clear up all remaining questions. In light of this, we kindly ask the reviewer to reconsider their rating.

---

> > > > ### Comment · Reviewer_Kyb9 · 2025-08-06
> > > >
> > > > I thank the authors for the response.
> > > >
> > > > - Granularity of Concepts: I acknowledge that image-level concepts and object-level concepts are generated differently. However, I disagree that the difference in generation process produces differently detailed concepts if the concept space are still the same, since the same concept can appear from both the object and image-level concepts ("pelican" in the Figure 6 example). To support this claim, there should be at least a study to show that the generated image-level concepts are generally a different subset of concepts from the object-level concepts.
> > > > - Modularity: This concern is addressed, but this clarification should be made in the paper draft as well, as the current phrasing is misleading.
> > > > - Grouping: I am not convinced that OCB's setup conveys groupings of concepts to the predictor. Could the authors clarify how the information of which concepts belongs to which object is conveyed to the predictor if the concepts are aggregated into a single concept vector as the input to the predictor? How is it able to "trace concepts to objects"?
> > > > - Object-Level Explanations: This question is addressed.

---

> > > > > ### Author Response · Authors · 2025-08-07
> > > > >
> > > > > > Granularity of Concepts
> > > > >
> > > > > There might be a small misunderstanding of how the concepts are generated in our setup:
> > > > > We use SPLiCE as base concept extractor. Given a vocabulary (which consists of 10000 words), Splice takes the input image and encodes it with CLIP. It compares this encoding with the text encoding of all words, resulting in a vector of 10000 similarities. Then it optimizes this vector for sparsity, so that only a small number of entries remain non-zero, while maintaining as high of a similarity to the original image vector. In practice, this results in around 10 to 15 non-zero concepts.
> > > > >
> > > > > For OCB, we first obtain such a sparse concept vector for the base image, and then additional sparse concept vectors for each detected object. Typically, we saw a relatively low overlap between the image and object-level concepts. Therefore, the object-level concepts add more detail to the resulting concept representation which is also an explanation for the improved performance.
> > > > >
> > > > > To illustrate this, we conducted a study showing the average number of non-zero concepts for the base image and when adding one or more objects:
> > > > >
> > > > >
> > > > > | |Base|+1 Object|+2 Objects|+3 Object|+5 Object|
> > > > > |-|-|-|-|-|-|
> > > > > |VOC|12.6|20.8|26.2|29.5|33.2|
> > > > > |COCO|14.1|22.1|28.2|32.4|37.3|
> > > > >
> > > > > The number of activated concepts increases with the number of objects, suggesting that each object contributes additional information to OCB's concept representation.
> > > > >
> > > > > We also note that adding more concepts through OCB is more effective than just allowing for more image-level concepts (without the object bias), as our evaluations in the main paper regarding sparsity of the concept space have shown.
> > > > >
> > > > >
> > > > > > Modularity
> > > > >
> > > > > We agree and updated this in the revised version of the manuscript.
> > > > >
> > > > > > Grouping
> > > > >
> > > > > We are sorry that the previous response was not clear: For the prediction the concept grouping is maintained by concat but not by the other aggregations. When object-level explanations are concerned, tracing the concepts back to the individual objects can also work with other aggregations if concept spaces do not fully overlap.
> > > > >
> > > > > We hope that these answers can clarify the remaining questions and that the reviewer takes these into account for their final decision.

---

> > > > > > ### Comment · Reviewer_Kyb9 · 2025-08-07
> > > > > >
> > > > > > I thank the authors for the clarifications.
> > > > > >
> > > > > > Granularity of concepts: This study indeed validates that the object-level concepts are additional concepts not present in the image-level concepts. Given that these results are included in final draft, this concern is addressed.
> > > > > >
> > > > > > Grouping: As the claim that OCB "allows models to distinguish which concepts belong together, tied to specific objects." was made during the rebuttal but not in the paper draft, as long as this overstated claim is not made in the final draft, this concern is addressed as well.
> > > > > >
> > > > > > I will increase my score accordingly.

---

### Official Review · Reviewer_YQGn · 2025-07-02

**Clarity:** 3
**Significance:** 2
**Originality:** 3
**Rating:** 4
**Confidence:** 3

**Summary:**

This paper proposes to modify concept bottleneck models (CBMs) by first performing generic object detection, followed by a cropping of each detected object and an extraction of an object-specific CBM for each object. The concept-based representation of the whole image and the object proposals are then aggregated before being used to solve a downstream task. For all steps, pretrained models are used: MaskRCNN or SAM for object proposals, Splice with CLIP for building the concept bottlenecks.
A new COCO-based dataset is proposed (COCOLogic) in which the classes correspond to specific combinations of objects.
The results show that using object-centric concept bottleneck leads to improved performance with respect to plain CBMs.

**Questions:**

- What is the behaviour of OCB and CBM for varying levels of sparsity?
- What is the performance with a pure object-centric representation, without including the whole image concept representation?
- If I got it well, crops with object proposals are directly given to the pretrained CLIP image encoder. It seems that the smaller the object proposals, the further from the CLIP training data. Did the authors investigate whether the size of the object proposals has an impact on performance?
- Although the concepts are obtained separately for every object, it seems that all aggregation procedures combine the concepts in a way that the decision module cannot know which concepts belong together (to the same object). Is there a reason for this?

**Ethical Concerns:**

["NO or VERY MINOR ethics concerns only"]

**Final Justification:**

The authors have addressed my main concerns. However, I concur with reviewer Kyb9 that some of the paper's claims are overstated. I lean slightly towards acceptance, although these issues need to be addressed first.

I quote from a discussion with Kyb9, with whom I agree that the conclusion may be misleading.

> Additionally, the authors wrote in the rebuttal,

>>    This difference is not merely about granularity; it provides explicit grouping of concepts by object, which is not available in global image-level approaches.
>>
>>    It allows models to distinguish which concepts belong together, tied to specific objects.
>>
>>    It facilitates compositional and interpretable reasoning, aligning more closely with how humans understand scenes.

> However, the concept groupings by object boundaries are not actually employed during the prediction. The concepts are aggregated but the groupings are not maintained (except for concat, but even there the groupings are not conveyed). As such, I find these claims in the rebuttals to also be unfounded.

**Limitations:**

There is a lack of discussion about several of the limitations of the approach, including computational considerations, the fact that the decision is taken over aggregated concepts (removing object information) and the connections between COCOLogic and some real tasks.

**Quality:**

2

**Strengths And Weaknesses:**

Strengths:
- Object-centric representations are a sensible choice for many tasks, and this work shows that extracting concepts from image crops obtained via object proposals can lead to improved performance on some downstream tasks.
- The approach is very simple and uses preexisting pretrained models, not requiring any training except for concept discovery from CLIP using Splice.
- The paper is clearly written and easy to follow.

Weaknesses:
Although I think the results are worth sharing, it is not clear to me whether they would qualify for NeurIPS:
- The only baseline is a regular CBM, with no alternatives, competitors or upper bounds considered. This makes the performance assessment relatively poor and does not put the results in their full context. A more competitive baseline that does not require additional forward passes would be to select the token embeddings (after a single forward pass) that correspond spatially to each object proposal.
- A more detailed evaluation of the “equal capacity” CBM would be required. After all, the sparsity of OCB is likely related to the number of object proposals (and their diversity). Does the “equal capacity” CBM follow the same trends? It would be important to see the sparsity vs performance curves for both approaches.
- The proposed COCOLogic, where the proposed model performs best, remains a rather artificial dataset. There is no discussion about how this would measure progress on some real task.
- There is no assessment nor discussion regarding computational considerations at inference time. Although the proposed approach requires very light training (just Splice), it seems to me it requires as many forward passes as there are object proposals.

---

> ### Author Rebuttal · Authors · 2025-07-31
>
> We thank the reviewer for their helpful suggestions and the appreciation of our results. We provide clarifications for the raised points below, starting with a general response regarding the novelty of our method followed by additional baselines.
>
> > Novelty
>
> We acknowledge that the individual components of our Object-Centric Bottleneck (OCB) framework, such as object proposal generation and concept classification, are not novel in isolation. However, the novelty of our work does not lie in proposing a new object detection mechanism, but in how we integrate object-centric representations into the Concept Bottleneck Model (CBM) framework, making it scalable and applicable to complex, real-world visual data.
>
> To the best of our knowledge, this is the first work that operationalizes object-centric concept representations within CBMs and applies them to real-world, multi-label classification tasks. Traditional CBMs rely on global, image-level features and often suffer from spurious or entangled concept associations [1], especially in cluttered scenes. In contrast, our object-centric bottleneck grounds concepts in specific, localized regions via pretrained object detectors, enabling finer-grained, interpretable, and more faithful concept-level reasoning.
>
> While this design may appear simple, bridging object-centric representations with CBMs in a real-world setting is non-trivial and has not been demonstrated before. Prior work exploring object-centric inductive biases has largely been limited to synthetic and single-class environments. Our contribution lies in extending these insights to realistic, diverse, and noisy data, removing the need for manual concept annotations, improving cross-dataset generalization, and demonstrating that object-level concept reasoning is viable and beneficial in practical scenarios.
>
> In summary, the strength and novelty of our approach lie in making CBMs more structured, scalable, and interpretable by rethinking their core representation—from flat, image-level concepts to grounded, object-level ones without requiring custom supervision or architectures.
>
>
>
> > Additional baselines.
>
> We agree that adding additional baselines strengthens the contextualization of our results which we highlight below:
>
> ### 1. Opaque Upper-Bound Model
> To quantify the performance potential of our concept encoder backbone (CLIP), we introduce a non-interpretable model that directly maps CLIP image embeddings to class predictions using a non-linear multi-layer perceptron (MLP) (2 layers). It serves as an upper bound on the performance that can be achieved using CLIP features alone. As expected, it outperforms interpretable CBM variants slightly, but comes at the cost of complete opacity, offering no meaningful explanations. OCB closes the performance gap between CBM and the opaque model, underscoring its value for transparent, high-performing concept reasoning.
>
> |               | Pascal-VOC         | COCO(h)           | COCO(l)         | SUN               | COCOLogic         |
> |---------------|--------------------|-------------------|------------------|--------------------|--------------------|
> | CBM | $82.42 \pm 0.01$ | $84.73 \pm 0.00$ | $59.19 \pm 0.00$ | $74.79 \pm 0.01$ | $58.84 \pm 0.09$ |
> | **OCB (RCNN)** | $85.75 \pm 0.01$ | $87.33 \pm 0.00$ | $64.12 \pm 0.00$ | $75.28 \pm 0.04$ | $68.84 \pm 0.11$ |
> | Upper Bound   | $89.60 \pm 0.05$   | $89.67 \pm 0.03$  | $68.20 \pm 0.13$ | $79.62 \pm 0.11$   | $65.95 \pm 1.00$   |
>
>
> Caption: Upper bound with frozen CLIP encodings and MLP classifier compared to CBM and OCB.
>
> ### 2. Coarse-to-Fine CBM (C2F)
> We further evaluate against the recent Coarse-to-Fine CBM [2], which introduces hierarchical, patch-based concept processing. Unlike many CBM variants that differ primarily in vocabulary, C2F contributes a distinct architectural inductive bias. To ensure a fair comparison focused on architecture rather than vocabulary, we ran C2F with the same LAION-based concept vocabulary of our OCB setup and compare to OCB without sparsity constraints. Since C2F only supports single-label classification, we evaluate it on the SUN397 and COCOLogic datasets. We observe that our OCB model consistently outperforms C2F under these conditions, highlighting the benefit of object-centric representations over patch-based hierarchies.
>
>
> |               | SUN                | COCOLogic         |
> |---------------|--------------------|--------------------|
> | C2F           | $61.28 \pm 1.44$                  | $66.27 \pm 1.09$   |
> | OCB (ours)    | $78.11 \pm 0.57$   | $68.63 \pm 0.42$   |
>
> Caption: Comparison to Coarse-to-Fine CBM (with LAION vocabulary)
>
> All results and discussions have been integrated into the main paper, and we thank the reviewers for this valuable suggestion.
>
>
> > Sparsity vs. Performance curves
>
> We next analyze the impact of the sparsity constraint on the base CBM and OCB (Pascal-VOC). We evaluate both models (Base CBM and OCB) with varying $\ell_1$-penalties, which controls the number of active concepts for image and concept-level encodigns. As reference, we also provide the result where both models are not constrained and generate activations for every concept.
>
> The results below show that both models improve as sparsity is relaxed. However, **OCB consistently outperforms the Base CBM across all sparsity levels**, indicating that object-centric representations yield more informative and efficient concepts.
>
> These trends confirm that the performance gain of OCB is not due to increased capacity, but rather to the stronger inductive bias of object-centric representations. We have added this experiment and discussion to the main paper.
>
> | Avg. # of non-zero Concepts  | 15 | 25 - 30 | 60 - 70  | 130 - 160   | |10000 (no sparsity) |
> |-|-|-|-|-|-|-|
> | Base CBM      | $82.42 \pm 0.01$   | $83.49 \pm 0.02$   | $84.07 \pm 0.01$   | $84.14 \pm 0.01$   | |$88.27 \pm 0.17$   |
> | OCB (ours)    |- | $85.75 \pm 0.01$   | $86.27 \pm 0.01$   | $86.35 \pm 0.00$ | |$90.56 \pm 0.02$   |
>
> > COCOLogic relevance
>
> While we agree that the COCOLogic dataset includes constructed labels, this is by design: COCOLogic is intended as a **proxy task for structured visual reasoning within single-class image classification**, i.e., to capture aspects such as multi-object composition, negation, and hierarchical categories while still using **real-world visual data**.
> For example, distinguishing a "personal transport scene" involving a person on a bicycle versus an image of a bicycle alone without a person is the kind of **entity-based and compositional reasoning** that object-centric models are meant to handle.
> We have added a discussion of this point in Section 4 of the revised manuscript to clarify its motivation and role in benchmarking progress toward **real-world reasoning tasks**, such as visual question answering and scene understanding.
>
> > Ablation: only object encodings
>
> We further observe that removing the image-level encodings from OCB leads to a substantial performance drop (Pascal VOC), confirming that **object-centric and global representations provide complementary information**.
> These results support a key message of our work: **both object-centric and global features are necessary** for achieving high predictive performance. We thank reviewer YQGn for this insightful suggestion.
>
> | | Full (Object + Image) | Only Image Encodings | Only Object Encodings |
> |-|-|-|-|
> | VOC | $85.75 \pm 0.01$ | $82.42 \pm 0.01$ | $77.48 \pm 0.03$ |
>
> > Limitations discussion.
>
> We agree that one limitation of our approach is the reduced computational efficiency, as multiple CLIP inference runs are necessary. However, as it is possible to batch the runs of multiple objects together, this should not be a major hinderance in practice. Nevertheless, we added this limitation to the discussion section of the paper.
>
> > Ablation: Impact object size.
>
> We appreciate the reviewer YQGn's observation and agree that very small object crops may fall outside the typical CLIP training distribution. We therefore evaluated OCB on Pascal VOC using different minimum object size thresholds (as a fraction of the image area). The results below show that performance **gradually degrades as smaller objects are excluded**, suggesting that **including smaller object proposals contributes positively**, despite the potential domain gap:
>
> | Minimum Size Threshold | 0.005| 0.01| 0.02 | 0.05|
> |-|-|-|-|-|
> | VOC Accuracy| $85.70 \pm 0.01$ | $85.59 \pm 0.01$ | $85.09 \pm 0.01$ | $84.59 \pm 0.02$ |
>
> > Aggregation methods object mapping?
>
> While four out of the five aggregation methods do lose the explicit mapping of concepts to individual objects, the concat method retains this information and offers a good trade-off—provided memory usage and a fixed object limit are not an issue. We discuss this in Section 5.3. Additionally, gradient-based techniques or attribution methods could potentially help disentangle which objects contribute to a decision, even when using pooled representations.
>
> We trust that these clarifications and additional results strengthen the clarity and completeness of our work, and we respectfully encourage the reviewer to reconsider their evaluation in light of these updates.
>
>
> [1] Marconato et al. "Not all neuro-symbolic concepts are created equal: Analysis and mitigation of reasoning shortcuts." NeurIPS, 2023.
>
> [2] Panousis et al. "Coarse-to-Fine Concept Bottleneck Models." NeurIPS, 2024.

---

### Official Review · Reviewer_gmMc · 2025-07-03

**Clarity:** 2
**Significance:** 2
**Originality:** 2
**Rating:** 4
**Confidence:** 3

**Summary:**

This work proposes a framework that combines concept-based models with pre-trained object-centric models to improve interpretability and performance for complex computer vision tasks. The authors propose three components to implement this: an object proposal module, a concept discovery module, and a predictor module. For implementation, their approach combines pre-trained object detectors (Mask R-CNN/SAM) with concept extractors (SpLiCE/CLIP), applies concept extraction to object crops, and trains linear classifiers on aggregated features. The authors evaluate OCB across multiple datasets and additionally introduce COCOLogic, a new benchmark dataset designed to evaluate the model’s ability to perform complex reasoning in a single-label classification setting.

**Questions:**

Please try to address my questions in the Weaknesses section.

**Ethical Concerns:**

["NO or VERY MINOR ethics concerns only"]

**Final Justification:**

Thanks to the authors for their response! Some of my concerns have been addressed. The additional baselines look like a good enhancement of the main contribution of this work. However, I do feel that a deeper analysis of the bias/error of the dependency models (e.g., CLIP and Mask-RCNN) would benefit future work in this direction. Accordingly, I have increased my score.

**Limitations:**

Yes

**Paper Formatting Concerns:**

No major concern.

**Quality:**

2

**Strengths And Weaknesses:**

Strengths
* This work tackles a limitation of current concept bottleneck models (not well-suited for complex tasks like multi-label classification due to image-level encoding) by extending to object-level encoding, which allows concept bottleneck models to handle more complex visual reasoning tasks including multi-label classification and logical reasoning visual tasks.
* The paper proposes three key components (i.e., object proposal, concept discovery, aggregation) and conducts various ablation studies to analyze the effectiveness of each component.
* The authors also introduces COCOLogic, a new dataset based on MSCOCO designed to evaluate models' ability to perform complex logical reasoning over natural visual content.

Weaknesses
* The novelty is a bit unclear. For instance, the 'From Image to Object' module appears similar to standard object detection pipelines with basic filtering.
* The aggregation mechanisms show no clear winner across datasets (e.g., Figure 4). How does this generalize to real-world applications?
* Given that the proposed approach heavily relies on pretrained models (Mask-RCNN, SpLiCE), and the authors mention in Section 5.4 that low-quality proposals can hurt performance, a more comprehensive analysis, such as showing and analyzing more failure cases, would be more insightful for future work on developing a better aggregation module.

---

> ### Author Rebuttal · Authors · 2025-07-31
>
> We thank the reviewer for their time and questions. We provide clarifications for the points that were unclear below.
>
> > Novelty is a bit unclear.
>
> We acknowledge that the individual components of our Object-Centric Bottleneck (OCB) framework, such as object proposal generation and concept classification, are not novel in isolation. However, the novelty of our work does not lie in proposing a new object detection mechanism, but in how we integrate object-centric representations into the Concept Bottleneck Model (CBM) framework, making it scalable and applicable to complex, real-world visual data.
>
> To the best of our knowledge, this is the first work that operationalizes object-centric concept representations within CBMs and applies them to real-world, multi-label classification tasks. Traditional CBMs rely on global, image-level features and often suffer from spurious or entangled concept associations [1], especially in cluttered scenes. In contrast, our object-centric bottleneck grounds concepts in specific, localized regions via pretrained object detectors, enabling finer-grained, interpretable, and more faithful concept-level reasoning.
>
> While this design may appear simple, bridging object-centric representations with CBMs in a real-world setting is non-trivial and has not been demonstrated before. Prior work exploring object-centric inductive biases has largely been limited to synthetic and single-class environments. Our contribution lies in extending these insights to realistic, diverse, and noisy data, removing the need for manual concept annotations, improving cross-dataset generalization, and demonstrating that object-level concept reasoning is viable and beneficial in practical scenarios.
>
> In summary, the strength and novelty of our approach lie in making CBMs more structured, scalable, and interpretable by rethinking their core representation—from flat, image-level concepts to grounded, object-level ones without requiring custom supervision or architectures.
>
>
> > "The aggregation methods show no clear winner accross datasets"
>
> We are somewhat puzzled the reviewer considers this as a weakness of the paper as it answers a practically relevant question, namely that the specific choice of aggregation method is often not critical. As discussed in Section 5.3, in the absence of dataset-specific knowledge or validation data, simple methods like *sum* or *max* generally perform best. More complex options like *concat* or *sum+count* can help, e.g., in terms of retaining object-mapping. Importantly, we further observe that on all object-based tasks, OCB outperforms the non object-based baseline **independent of the aggregation method** used.
>
>
> > Reliance on pre-trained models
>
> We agree that the reliance on pretrained models such as Mask-RCNN and SpLiCE introduces potential failure points, especially when their outputs are of low quality (cf. Section 5.4). However, their usage also brings a major advantage: it enables concept extraction at the object level for real-world data without the need for task-specific, large-scale annotation or supervision. This addresses a well-known limitation of prior CBM approaches, which often require carefully curated concept annotations or extensive domain-specific training [2, 3].
>
>
> > More comprehensive analysis
>
> We agree a failure-case analysis is beneficial and have added such a section to the paper. As we cannot add images and plots here, we describe some of the failure cases of the proposal generation that we encountered.
>
> **Overlapping proposals:** Sometimes it can happen that the object proposal model generates multiple proposals from the same object, despite filtering them for a low overlapping IOU. In this case, the same object appears in the concept space multiple times, which can lead to incorrect results if the exact number of concepts is relevant.
>
> **Detecting irrelevant objects:** If some objects are inherently not relevant for the task, detecting them and adding them to the concept space can introduce spurious correlations. For instance, in the "plane interior" class of the SUN dataset, most samples also contain humans, which are detected as objects. This can lead the model to incorrectly associate the presence of humans with the "plane interior" label.
>
>
> > Additional baseline results
>
> We additionally wish to highlight novel baseline and ablation evaluations resulting from other reviews and strengthen the findings of our work.
>
> ### 1. Opaque Upper-Bound Model
> To quantify the performance potential of our concept encoder backbone (CLIP), we introduce a non-interpretable model that directly maps CLIP image embeddings to class predictions using a non-linear multi-layer perceptron (MLP) (2 layers). It serves as an upper bound on the performance that can be achieved using CLIP features alone. As expected, it outperforms interpretable CBM variants slightly, but comes at the cost of complete opacity, offering no meaningful explanations. OCB closes the performance gap between CBM and the opaque model, underscoring its value for transparent, high-performing concept reasoning.
>
> |               | Pascal-VOC         | COCO(h)           | COCO(l)         | SUN               | COCOLogic         |
> |---------------|--------------------|-------------------|------------------|--------------------|--------------------|
> | CBM | $82.42 \pm 0.01$ | $84.73 \pm 0.00$ | $59.19 \pm 0.00$ | $74.79 \pm 0.01$ | $58.84 \pm 0.09$ |
> | **OCB (RCNN)** | $85.75 \pm 0.01$ | $87.33 \pm 0.00$ | $64.12 \pm 0.00$ | $75.28 \pm 0.04$ | $68.84 \pm 0.11$ |
> | Upper Bound   | $89.60 \pm 0.05$   | $89.67 \pm 0.03$  | $68.20 \pm 0.13$ | $79.62 \pm 0.11$   | $65.95 \pm 1.00$   |
>
>
> Caption: Upper bound with frozen CLIP encodings and MLP classifier compared to CBM and OCB.
>
> ### 2. Coarse-to-Fine CBM (C2F)
> We further evaluate against the recent Coarse-to-Fine CBM [4], which introduces hierarchical, patch-based concept processing. Unlike many CBM variants that differ primarily in vocabulary, C2F contributes a distinct architectural inductive bias. To ensure a fair comparison focused on architecture rather than vocabulary, we ran C2F with the same LAION-based concept vocabulary of our OCB setup and compare to OCB without sparsity constraints. Since C2F only supports single-label classification, we evaluate it on the SUN397 and COCOLogic datasets. We observe that our OCB model consistently outperforms C2F under these conditions, highlighting the benefit of object-centric representations over patch-based hierarchies.
>
>
> |               | SUN                | COCOLogic         |
> |---------------|--------------------|--------------------|
> | C2F           | $61.28 \pm 1.44$                  | $66.27 \pm 1.09$   |
> | OCB (ours)    | $78.11 \pm 0.57$   | $68.63 \pm 0.42$   |
>
> Caption: Comparison to Coarse-to-Fine CBM (with LAION vocabulary).
>
>
> ### Sparsity Analysis
>
> We next analyze the impact of the sparsity constraint on the base CBM and OCB (Pascal-VOC). We evaluate both models (Base CBM and OCB) with varying $\ell_1$-penalties, which controls the number of active concepts for image and concept-level encodigns. As reference, we also provide the result where both models are not constrained and generate activations for every concept.
>
> The results below show that both models improve as sparsity is relaxed. However, **OCB consistently outperforms the Base CBM across all sparsity levels**, indicating that object-centric representations yield more informative and efficient concepts.
>
> These trends confirm that the performance gain of OCB is not due to increased capacity, but rather to the stronger inductive bias of object-centric representations. We have added this experiment and discussion to the main paper.
>
> | Avg. # of non-zero Concepts  | 15 | 25 - 30 | 60 - 70  | 130 - 160   | |10000 (no sparsity) |
> |---------------|--------------------|--------------------|--------------------|--------------------| -|--------------------|
> | Base CBM      | $82.42 \pm 0.01$   | $83.49 \pm 0.02$   | $84.07 \pm 0.01$   | $84.14 \pm 0.01$   | |$88.27 \pm 0.17$   |
> | OCB (ours)    |- | $85.75 \pm 0.01$   | $86.27 \pm 0.01$   | $86.35 \pm 0.00$                  | |$90.56 \pm 0.02$   |
>
>
> ### OCB: Contribution of Object vs. Image-Level Encodings
>
> Finally, we observe that removing the image-level encodings from OCB leads to a substantial performance drop (Pascal VOC), confirming that **object-centric and global representations provide complementary information**.
> These results support a key message of our work: **both object-centric and global features are necessary** for achieving high predictive performance. We thank reviewer YQGn for this insightful suggestion.
>
> |               | Full (Object + Image) | Only Image Encodings | Only Object Encodings |
> |---------------|------------------------|------------------------|------------------------|
> | VOC           | $85.75 \pm 0.01$       | $82.42 \pm 0.01$       | $77.48 \pm 0.03$       |
>
> All novel results and discussions have been integrated into the main paper.
>
> We hope that these clarifications and additional results strengthen the clarity and completeness of our work, and we respectfully encourage the reviewer to reconsider their evaluation in light of these updates.
>
>
> [1] Marconato et al. "Not all neuro-symbolic concepts are created equal: Analysis and mitigation of reasoning shortcuts." NeurIPS, 2023.
>
> [2] Koh, Pang Wei, et al. "Concept bottleneck models." International conference on machine learning. PMLR, 2020.
>
> [3] Yang, Yue, et al. "Language in a bottle: Language model guided concept bottlenecks for interpretable image classification." CVPR, 2023.
>
> [4] Panousis et al. *"Coarse-to-Fine Concept Bottleneck Models."* NeurIPS, 2024.

---

### Decision · Program_Chairs · 2025-09-17

**Decision:**

Accept (poster)

**Comment:**

This paper introduces Object-Centric Concept Bottleneck, an interpretable concept-based model for image classification which emphasizes the model’s ability to distinguish multiple different object types in the same image. First, a pre-trained “object proposal” model is used to identify candidate concepts and bounding boxes. Then, after some post processing tricks (removing duplicates or low-confidence concepts, and cropping and resizing), the newly formed images are aggregated, together with the full image, and fed through a subsequent “concept discovery” module (which is also pre-trained), which outputs a feature representation formed by a score for each of a finite number $C$ of concepts. Lastly, a trainable linear layer is employed on the concept layer to produce the final classification. In experiments on several datasets, the authors demonstrate improved performance compared to an interpretable concept-based model baseline. In addition, a **new dataset, COCOLogic** is introduced which is based on the COCO dataset but creates artificial classes which consist in logical conjunctions of the individual objects present in the image (further details are available in Section D and in table 2 on page 16). For instance, there is a class corresponding to the presence of exactly one of “dog” and “car” (dog XOR car). Similarly, the method outperforms the simple baseline in this case.


Whilst some of the reviewers were initially skeptical regarding the novelty (e.g. Rev. gmMc) and the lack of comparison with more (not-necessarily interpretable) baselines (Rev. YQGn), **all reviewers eventually agreed** that the reasons to accept outweigh the reasons to reject. Therefore, I am happy to recommend **acceptance**. However, I strongly urge the authors to work hard on the camera-ready revision to address both the reviewers’ and my concerns.


In particular, more baselines should be added, even if the non-interpretable ones outperform the current method. In addition, some additional details are needed to make the paper clearer.  For instance, the fact that both the object proposal and concept discovery modules are pretrained is only mentioned in lines 207 to 210 at the end of section 3, which makes the paper less friendly to readers unfamiliar with the current craze for pre-trained models and fine-tuning.  (I appreciate reviewer gmMc’s clarification that the modules are typically Mask R-CNN/SAM and SpLiCE/CLIP respectively). Other clarifications on the experimental settings are required in the camera-ready version (including the mode detailed explanation in the rebuttal comment https://openreview.net/forum?id=9lhijvd0fs&noteId=gDfk8Ftwqi).

Lastly, Reviewers Kyb9 and YQGn both have concerns regarding the meaning of the concepts and groupings generated by the two main pre-trained components. Neither reviewer was fully satisfied by the authors’ answer, though neither reviewer thought that this justified rejecting the paper. I also agree that the **interpretability of the last concept layer is imperfect**: considering the introduced COCOLogic dataset, since XOR relationships are not satisfactorily representable by linear maps, it must mean that either the concept classes at the last pre-trained layer already contain composite features, or *the performance of the introduced method is actually poor on this dataset (on the XOR-type classes) in an absolute sense* and it only outperforms a weak baseline also based on a linear combination of concepts. I suspect the latter is true, in which case the authors should also **investigate the use of a full 1-3 layer MLP at the final prediction layer** and **evaluate the class-wise performance**.

However, I still argue for acceptance as all the reviewers are positive and I also find that the introduction of datasets such as COCOLogic to be a strong and creative contribution which takes the community in the right direction of explorative research into new tasks and interpretability (as opposed to obsessing over infinitesimal performance improvements on well-established datasets).


**Minor typos**:

Line 175 “for in input” should be “for an input”.